# INVERSIONGNN: A DUAL PATH NETWORK FOR MULTI-PROPERTY MOLECULAR OPTIMIZATION

**Yifan Niu[1], Ziqi Gao[1,2], Tingyang Xu[3], Yang Liu[1,2], Yatao Bian[4], Yu Rong[3], Junzhou Huang[5], Jia Li[1,2]***
[1]The Hong Kong University of Science and Technology (Guangzhou)
[2]The Hong Kong University of Science and Technology
[3]DAMO Academy, Alibaba Group  [4]Tencent AI Lab  [5]University of Texas, Arlington

## ABSTRACT

Exploring chemical space to find novel molecules that simultaneously satisfy multiple properties is crucial in drug discovery. However, existing methods often struggle with trading off multiple properties due to the conflicting or correlated nature of chemical properties. To tackle this issue, we introduce InversionGNN framework, an effective yet sample-efficient *dual-path* graph neural network (GNN) for multi-objective drug discovery. In the **direct prediction path** of InversionGNN, we train the model for multi-property prediction to acquire knowledge of the optimal combination of functional groups. Then the learned chemical knowledge helps the **inversion generation path** to generate molecules with required properties. In order to decode the complex knowledge of multiple properties in the inversion path, we propose a gradient-based Pareto search method to balance conflicting properties and generate Pareto optimal molecules. Additionally, InversionGNN is able to search the full Pareto front approximately in discrete chemical space. Comprehensive experimental evaluations show that InversionGNN is both effective and sample-efficient in various discrete multi-objective settings including drug discovery. The code is available at https://github.com/ivanniu/InversionGNN.

## 1 INTRODUCTION

Molecular optimization refers to the process of systematically modifying the structure of a given molecule to enhance its properties in practical drug discovery. It combines known chemical knowledge with innovative exploration to discover and develop unknown high-performance molecules. Molecular optimization is challenging as it usually involves reasoning about multiple, often conflicting or correlated, objectives (Fromer & Coley, 2023). For example, for a new drug to be successful, it must simultaneously be potent, bioavailable, safe, and synthesizable (Dara et al., 2022). Generally, these objectives often exhibit implicit relationships, which can be either conflicting or correlated, rather than being independent (Jain et al., 2023). For example, molecules that are effective against a target may also have detrimental effects on humans.

Although several drug discovery models have been proposed to tackle Multi-Objective Molecular Optimization (MOMO), most of them do not make full use of acquired chemical knowledge, such as how the combination of substructures affects chemical properties. More specifically, they merely employ a pretrained chemical property predictor as the discriminator, filtering high-performance molecules based on the predicted scores (Nigam et al., 2020; Xie et al., 2021; Brown et al., 2019). Additionally, most studies neglect the conflicting or correlated relationships among chemical properties and simply use a predefined Linear Scalarization function (e.g., mean) to perform a weighted summation of the losses (Jin et al., 2018; Wang et al., 2023; De Cao & Kipf, 2018; Shi et al., 2020; Liu et al., 2021; Fu et al., 2022; Zhou et al., 2019; Jin et al., 2020b; Jain et al., 2023; Xie et al., 2021). However, Linear Scalarization often results in biased solutions and leaves certain areas of objective space unexplored, which has been mathematically analyzed by Boyd et al. (2004). Abbasi et al. (2022) employs multi-objective Genetic Algorithms to tackle MOMO, but it is computationally expensive and requires a large number of Oracle calls. Recently, Jain et al. (2023) and Zhu et al.

---
*Correspondence to: Jia Li (`jialee@ust.hk`).

(2024) adopt the multi-objective Bayesian Optimization to address MOMO. Nevertheless, both of them suffer from high computational costs and struggle with high-dimensional optimization.

To design an effective yet sample-efficient model for MOMO, we identify two key challenges:

1. ***How to make full use of acquired chemical knowledge to facilitate molecular optimization?*** To tackle this, we introduce a dual-path graph neural network (GNN) (Kipf & Welling, 2016; Xu et al., 2019; Velickovic et al., 2018; Gilmer et al., 2017; Gao et al., 2024b; Liu et al., 2024; Gao et al., 2023; Li et al., 2024; 2025) to incorporate complicated chemical knowledge and perform molecular property prediction in the *direct prediction path*. Unlike existing methods that regard it as a discriminator, in the *inversion generation path*, we leverage the gradient w.r.t the molecule structure to optimize the molecular graph. The mapping of chemical structure to properties, learned through the direct path, can effectively guide the editing of chemical structure to a base molecule in the molecular optimization process.

2. ***How to dealing with the conflicting or correlated properties in the inversion generation path?*** In our dual-path GNN model, the direct path easily learns each property distribution using different classification heads. However, in the inversion generation process, it is challenging to apply multiple complicated property constraints to one single molecule. For example, given two conflicting properties, enhancing one property will weaken the other property. To capture all possible trade-offs among conflicting or correlated properties, we introduce the gradient-based Pareto optimization technique, which is designed to balance multiple conflicting or correlated objectives (Zhou et al., 2023; Ju et al., 2022; Liu et al., 2022). Instead of directly deploying existing Pareto methods, we adopt the *relaxation* technique to adapt the continuous Pareto optimization to the *discrete* chemical space. We provide a convergence analysis demonstrating that our approach, with our relaxation, still converges to the Pareto optimal solutions approximately within only a few iterations. This means our method can *effectively* increase the likelihood of successfully generating high-quality molecules with *sample efficiency*. Our key contributions are summarized below:

- We propose a novel dual-path InversionGNN for multi-objective molecular optimization, which leverages the acquired property prediction knowledge to facilitate molecular optimization. It is effective and sample-efficient.

- To capture trade-offs among conflicting or correlated properties, we adpot relaxation technique that adapts gradient-based Pareto optimization to the discrete chemical space.

- We empirically verify that InversionGNN is both effective and sample-efficient in various real-world discrete multi-objective settings including drug discovery.

## 2 RELATED WORK

**Molecular Optimization.** Recent years have witnessed the success of applying deep generative models and molecular graph representation learning in drug discovery. Most of the existing works can be categorized into two classes: Constrained Generative Model (CGM) and Combinatorial Optimization (CO) algorithm. CGMs model the molecular distribution with deep generative networks such as VAE (Gómez-Bombarelli et al., 2018; Liu et al., 2018; Jin et al., 2018; 2019; Skalic et al., 2019; Fu et al., 2020; Griffiths & Hernández-Lobato, 2020; Wang et al., 2023), GAN (Guimaraes et al., 2017; De Cao & Kipf, 2018; Abbasi et al., 2022), Flow (Shi et al., 2020), Energy (Liu et al., 2021) and Diffusion-based model (Lee et al., 2023), projecting input molecules into a latent space. However, obtaining the ideal smooth and discriminative latent space has proven to be a challenge in practice (Brown et al., 2019; Huang et al., 2021; Gao et al., 2025). Another research line based on CO directly searches for desired molecules in the explicit discrete space, e.g., Reinforcement Learning (You et al., 2018; Ståhl et al., 2019; Zhou et al., 2019; Jin et al., 2020b; Gottipati et al., 2020; Gao et al., 2024a; Jain et al., 2023; Popova et al., 2018; Jin et al., 2020a), Evolutionary Algorithms (Jensen, 2019; Nigam et al., 2020; Chen et al., 2021), Markov Chain Monte Carlo (Xie et al., 2021; Fu et al., 2021), Tree Search (Fu et al., 2022) and Bayesian Optimization (Korovina et al., 2020; Moss et al., 2020). CO algorithms require massive numbers of Oracle calls, which is computationally inefficient during the inference time. However, they are still challenged in dealing with conflicting or correlated properties.

**Gradient-Based Pareto Optimization.** The Pareto optimal solution is highly valuable for multi-objective optimization since identifying solutions that simultaneously maximize all objectives is often impractical. In order to efficiently find Pareto optimal solutions, MGDA (Désidéri, 2012) is proposed

to identify Pareto optimal solutions for low-dimensional data. Sener & Koltun (2018) extend MGDA to high-dimensional multi-objective scenarios. Subsequently, several efficient Pareto optimization methods (Lin et al., 2019; Zhang & Golovin, 2020; Ma et al., 2020; Mahapatra & Rajan, 2020) have been proposed to explore the Pareto set, due to the fact that MGDA cannot find Pareto optimal solutions specified by exact objective preference. However, most efforts of Pareto optimization focus on continuous parameter space, ignoring the complex discrete chemical space.

## 3 PRELIMINARIES

### 3.1 PARETO OPTIMALITY

In this work, we consider $m$ tasks described by $f(\boldsymbol{x}) := [f_i(\boldsymbol{x})]$, where each $f_i(\boldsymbol{x}), i \in [m]$ represents the performance of the $i$-th task to be maximized. Given an desired target $\boldsymbol{y} \in \mathbb{R}^m$, we set a non-negative objective function $\mathcal{L}(f(\boldsymbol{x}), \boldsymbol{y}) = [l_1, \ldots, l_m]^\mathsf{T}$, where $l_i$ for $i \in [m]$ is the objective function of the $i$-th task. Hence, maximizing the performance $f(\boldsymbol{x})$ is equivalent to minimizing the objective function. For any two points $\boldsymbol{x}, \boldsymbol{x}' \in \mathbb{R}^n$, $\boldsymbol{x}$ dominates $\boldsymbol{x}'$, denoted by $\mathcal{L}^{\boldsymbol{x}'} \succeq \mathcal{L}^{\boldsymbol{x}}$, implies $l_i^{\boldsymbol{x}'} - l_i^{\boldsymbol{x}} \geq 0, \forall i \in [m]$. A point $\boldsymbol{x}^*$ is said to be **Pareto optimal** if $\boldsymbol{x}^*$ is not dominated by any other points in $\mathbb{R}^n$. The set of all Pareto optimal solutions is denoted by $\mathcal{P}$. The set of multi-objective values of the Pareto optimal solutions is called **Pareto front**, denoted by $\mathcal{F}$. In Multi-Objective Optimization, the ideal goal is to identify a set of Pareto solutions that cover all the possible trade-offs among objectives. For a formal definition of Pareto concept, please refer to Appendix A.

### 3.2 DIFFERENTIABLE SCAFFOLDING TREES

A scaffolding tree (Jin et al., 2018), $\mathcal{T}_{\boldsymbol{x}}$, is a spanning tree whose nodes are substructures. It is a high-level representation of molecular graph $\boldsymbol{x} \in \mathcal{X}$. For a scaffolding tree with $K$ nodes and substructure set $\mathcal{S}$, it is represented by $\mathcal{T}_{\boldsymbol{x}} = \{\mathbf{N}, \mathbf{A}, \mathbf{w}\}$: (i) node indicator matrix defined by $\mathbf{N} \in \{0, 1\}^{K \times |S|}$, and each row of $N$ is a one-hot vector, indicating the substructure of the node; (ii) adjacency matrix denoted by $\mathbf{A} \in \{0, 1\}^{K \times K}$, where $\mathbf{A}_{ij} = 1$ indicates the $i$-th node and the $j$-th node are connected while $0$ indicates unconnected; and (iii) node weight vector $\mathbf{w} = [1, \ldots, 1]^\top \in \mathbb{R}^K$, indicates the $K$ nodes are equally weighted. Fu et al. (2022) add a virtual expansion node set $\mathcal{V}_{expand} = \{u_v \mid v \in \mathcal{V}_{\mathcal{T}_{\boldsymbol{x}}}\}, |\mathcal{V}_{expand}| = K_{expand} = K$ to the scaffolding tree for structure modification. $\mathcal{T}_{\boldsymbol{x}}$ can be converted to a $K + K_{expand}$ nodes differentiable scaffolding tree $\widetilde{\mathcal{T}}_{\boldsymbol{x}} = \{\widetilde{\mathbf{N}}, \widetilde{\mathbf{A}}, \widetilde{\mathbf{w}}\}$. The differentiable scaffolding tree facilitates the substructure addition, deletion, and replacement.

## 4 METHOD

In this study, we explore an InversionGNN framework to address the problem of (1) finding Pareto optimal molecules conditioned on desired weight and (2) finding diverse Pareto optimal molecules with all possible trade-offs. We illustrate the pipeline of InversionGNN in Figure 1:

- **A Dual-Path Network: InversionGNN.** We introduce a dual-path graph neural network (GNN) to incorporate complicated chemical knowledge in the *direct prediction path* (Section 4.2). In the *inversion generation path*, we leverage the gradient w.r.t the molecule structure to guide the molecular optimization process.

- **Gradient-Based Pareto Inversion.** In order to inverse the complicated multi-property knowledge, we relax the discrete molecule optimization into a locally differentiable Pareto optimization problem. We reorganize the gradients into a non-dominating gradient (Section 4.3).

### 4.1 PROBLEM FORMULATION

**Definition 4.1** (Multi-Objective Molecular Optimization (MOMO)). Given the Chemical Space $\mathcal{X}$, Oracle function $f(\boldsymbol{x})$, objective function $\mathcal{L}$, the target property score $\boldsymbol{y} \in \mathbb{R}^m$ of $m$ properties, the goal of MOMO is to find candidate molecules $\boldsymbol{x}^* \in \mathcal{X}$ that minimize all objectives:

$$\boldsymbol{x}^* = \arg\min_{\boldsymbol{x} \in \mathcal{X}} \mathcal{L}(f(\boldsymbol{x}), \boldsymbol{y}). \tag{1}$$

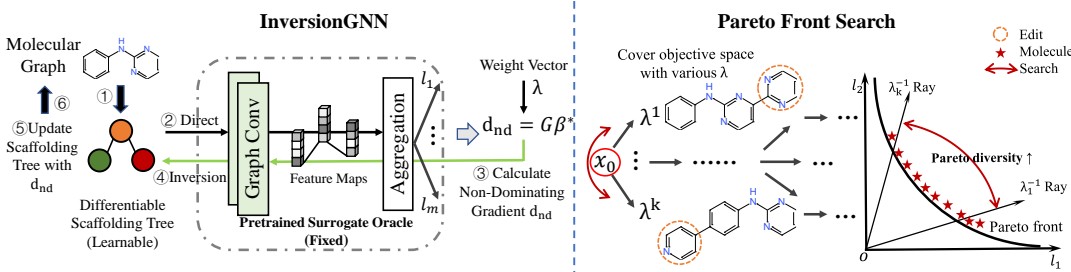

Figure 1: (1) **InversionGNN**. A surrogate Oracle GNN is trained to incorporate complicated chemical knowledge. In the direct prediction path, a molecule $\boldsymbol{x}^t$ is fed to the GNN to obtain the objective function at the $t$-th iteration. In the inversion path, we calculate the non-dominating gradient to find local Pareto-optimal molecules $\mathcal{P}^t$ conditioned on given weight vector $\boldsymbol{\lambda}$. (2) **Pareto Front Search**. Exploring the full Pareto front approximately with various weight vectors, improving the Pareto diversity of generated molecules.

When objectives $[l_1, \ldots, l_m]$ are conflicting, there is no single $\boldsymbol{x}^*$ which simultaneously maximizes all objectives. Consequently, multi-objective optimization adopts the concept of *Pareto optimality*, which describes a set of solutions $\mathcal{P}$ that provide optimal trade-offs among the objectives.

## 4.2 A DUAL-PATH NETWORK: INVERSIONGNN

In this section, we introduce a dual-path network, InversionGNN, consists of a *direct prediction path* and a *inversion generation path* for multi-objective molecular optimization.

**Direct Prediction Path.** We aim to develop a pretrained GNN $f(\boldsymbol{x}; \theta)$ to capture knowledge from the ground truth Oracle $\mathcal{O}$. We imitate the Oracle function $\mathcal{O}$ using a multi-head architecture to individually predict $m$ property scores $\widehat{\boldsymbol{y}} \in \mathbb{R}^m$:

$$\widehat{\boldsymbol{y}} = f(\boldsymbol{x}; \theta) \approx [\mathcal{O}_1(\boldsymbol{x}), \mathcal{O}_2(\boldsymbol{x}), \cdots, \mathcal{O}_m(\boldsymbol{x})]^\mathsf{T} = \boldsymbol{y}, \tag{2}$$

where $\theta$ is the learnable parameters. We adopt the Graph Convolutional Network (GCN) (Kipf & Welling, 2016) to extract the representations of molecules. The updating rule for the $l$-th layer is $H^{(l)} = \text{RELU}\left(B^{(l)} + A\left(H^{(l-1)}U^{(l)}\right)\right)$, where $B^{(l)} \in \mathbb{R}^{K \times d}/U^{(l)} \in \mathbb{R}^{d \times d}$ are bias/weight parameters and $A$ is adjacency matrix. We leverage the weighted average as the readout function of the last layer's node embeddings, followed by multi-head MLP to yield the prediction of $m$ properties $\widehat{\boldsymbol{y}} = \text{MLP}\left(\frac{1}{\sum_{k=1}^{K} w_k} \sum_{k=1}^{K} w_k H_k^{(L)}\right)$. We train the model by minimizing the discrepancy between the prediction $\widehat{\boldsymbol{y}}$ and the ground truth $\boldsymbol{y}$:

$$\theta^* = \arg \min_\theta \mathcal{L}\left(\boldsymbol{y}, \widehat{\boldsymbol{y}}\right), \tag{3}$$

where $\mathcal{L}$ is the loss function, e.g. binary cross entropy. The parameters of the surrogate Oracle model are pretrained at once and *freezed* in the inversion generation path.

**Inversion Generation Path.** Instead of using the GNN model as a simple predictor, we decode the stored chemical knowledge in the GNN model and use its gradient w.r.t the input molecule to guide molecular optimization. Let $g_i = \nabla l_i$ represent the gradient of the $i$-th property objective function. Consequently, we obtain $G = \nabla \mathcal{L} = [g_1, \ldots, g_m]$ by back-propagating the derivatives from the target properties. In order to ensure the molecular learnable, InversionGNN requires a differentiable molecular representation in chemical space. Compared to the widely used non-differentiable discrete scaffolding tree $\mathcal{T}_{\boldsymbol{x}}$, the currently only available differentiable representation is the differentiable scaffolding tree proposed by Fu et al. (2022). Hence, we employ the differentiable scaffolding tree, denoted as $\widetilde{\mathcal{T}}_{\boldsymbol{x}}$, as the representation of molecular $\boldsymbol{x}$ in InversionGNN.

Compared to InversionGNN, vanilla GCN (Kipf & Welling, 2016; Liu et al., 2023) is often used for prediction tasks, and its inference prediction process is the direct prediction path in InversionGNN. InversionGNN has the same structure as vanilla GCN but has a different computation process. In contrast, InversionGNN contains an additional inversion path for generation tasks, which allows the inverse of the gradients to the input molecules.

---

**Algorithm 1:** InversionGNN

---

**Input:** Input molecule $\boldsymbol{x}^0 \in \mathcal{X}$, weight vector $\boldsymbol{\lambda} \in \mathbb{R}^m$, and step size $\eta > 0$.
**Output:** Generated Molecule $\boldsymbol{x}^T$.

1   Initialization.
2   Train surrogate Oracle according to Eq. 3.
3   **for** $t = 1, \ldots, T$ **do**
4      Convert molecule $\boldsymbol{x}^t$ to differentiable scaffolding tree $\widetilde{\mathcal{T}}_{\boldsymbol{x}^t}^1$;
5      **for** $k = 1, \ldots, K$ **do**
6         Compute gradients of target objectives w.r.t. $\widetilde{\mathcal{T}}_{\boldsymbol{x}^t}^k$: $G = \nabla\mathcal{L} = [g_1, \ldots, g_m]$;
7         Determine $\boldsymbol{\beta}^*$ by solving QP problem as Eq. 5;
8         Calculate non-dominating gradient $d_{nd} = G\boldsymbol{\beta}^*$;
9         Update the differentiable scaffolding tree: $\widetilde{\mathcal{T}}_{\boldsymbol{x}^t}^{k+1} = \widetilde{\mathcal{T}}_{\boldsymbol{x}^t}^k - \eta d_{nd}$;
10      **end**
11      Sample discrete $\mathcal{T}_{\boldsymbol{x}^{t+1}}$ from continuous $\widetilde{\mathcal{T}}_{\boldsymbol{x}^t}^K$ and assemble it to molecule $\boldsymbol{x}^{t+1}$.
12   **end**

---

## 4.3   Gradient-Based Pareto Inversion

In this section, to capture possible trade-offs among conflicting or correlated properties, we adopt the relaxation technique and reformulate the discrete molecule Pareto optimization into a locally differentiable problem. At the $t$-th iteration, given one molecule $\boldsymbol{x}^t$, we aim to find local Pareto optimal molecules set $\mathcal{P}^t$ from the neighborhood set $\mathcal{N}(\boldsymbol{x}^t)$:

$$\boldsymbol{x}^{t+1} \in \mathcal{P}^t \subseteq \mathcal{N}(\boldsymbol{x}^t) \tag{4}$$

where $\mathcal{N}(\boldsymbol{x}^t)$ is the set of all the possible molecules obtained by (1) imposing one local editing operation (expand, remove or replace one substructure) to scaffolding tree and (2) assembling the edited trees into molecules.

**Identifying the Non-Dominating Gradient.** To approach the Pareto front, Désidéri (2012) demonstrated that the descent direction $d$ can be found within the convex hull of the gradients, i.e., $d \in \mathcal{CH}_{\boldsymbol{x}} := \{G\boldsymbol{\beta}\}$, where $\boldsymbol{\beta} \in \mathcal{S}^m$ belongs to the $m$-dimensional simplex. In order to identify the *Non-Dominating Descent Direction* $d_{nd} = G\boldsymbol{\beta}^*$, inspired by continuous Pareto optimization (Mahapatra & Rajan, 2020), we solve the following *Quadratic Programming* (QP) problem:

$$\boldsymbol{\beta}^* = \underset{\|\boldsymbol{\beta}\|_1 \leqslant 1}{\arg\min} \left\| G^\top G\boldsymbol{\beta} - \mathbf{a} \right\|^2$$

$$\text{s.t. } \boldsymbol{\beta}^\top G^\top g_j \geqslant 0 \quad \forall j \in \mathrm{J} = \left\{ \begin{array}{ll} \mathrm{J}^* & \mathrm{KL}\left(\mathcal{L} \odot \boldsymbol{\lambda} | \mathbf{1}\right) \leqslant \epsilon \\ [m] & \mathrm{KL}\left(\mathcal{L} \odot \boldsymbol{\lambda} | \mathbf{1}\right) > \epsilon \end{array} \right. , \tag{5}$$

$$\text{where} \quad \mathrm{J}^* = \left\{ j \in [m] \mid j = \arg\max_{j' \in [m]} l_{j'}\lambda_{j'} \right\},$$

$\mathbf{a}$ is the anchoring direction (Mahapatra & Rajan, 2020), and $\boldsymbol{\lambda} \in \mathbb{R}^m$ is a predefined weight vector that indicates the importance of each property. Then, we calculate the non-dominating direction $d_{nd} = G\boldsymbol{\beta}^*$ and update the differentiable scaffolding tree with $\widetilde{\mathcal{T}}_{\boldsymbol{x}} = \widetilde{\mathcal{T}}_{\boldsymbol{x}} - \eta d_{nd}$. Therefore, we can yield a solution $\widetilde{\mathcal{T}}_{\boldsymbol{x}^t}^*$ that is not dominated by $\widetilde{\mathcal{T}}_{\boldsymbol{x}^t}$ in its neighborhood set $\mathcal{N}(\widetilde{\mathcal{T}}_{\boldsymbol{x}^t})$.

**Molecule Search in Discrete Chemical Space.** At the $t$-th iteration, we begin with a molecule $\boldsymbol{x}^t$ and convert it to differentiable scaffolding tree $\widetilde{\mathcal{T}}_{\boldsymbol{x}^t}$. Subsequently, we identify the local Pareto optimal solution $\widetilde{\mathcal{T}}_{\boldsymbol{x}^t}^*$ within the neighborhood set $\mathcal{N}(\widetilde{\mathcal{T}}_{\boldsymbol{x}^t})$ by performing $K$ rounds of gradient descent against the non-dominating direction $d_{nd}$. From $\widetilde{\mathcal{T}}_{\boldsymbol{x}^t}^*$, we can sample the discrete scaffolding tree $\mathcal{T}_{\boldsymbol{x}^t}^*$ and assemble it to molecules, denoted as $\boldsymbol{x}^{t+1}$ in the following iteration. Our proposed InversionGNN is summarized in Algorithm 1.

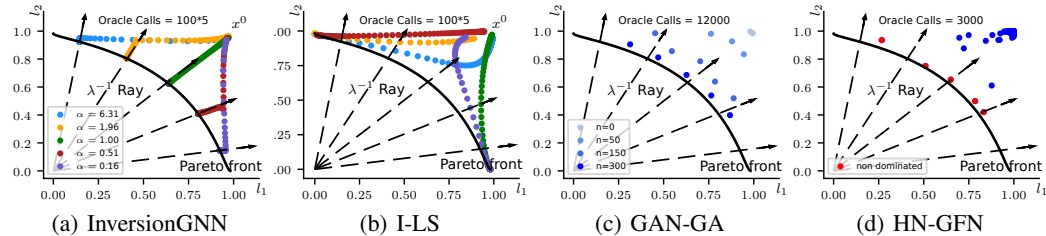

Figure 2: Pareto front (black solid curve) for two loss functions $l_1,l_2$ and solutions (circles) and Oracle calls (computational cost) for different weights $\alpha = \frac{\lambda_1}{\lambda_2}$ (dashed rays). The weight $\lambda$ conditioned Pareto optimal solution is the intersection points between the Pareto front and $\lambda^{-1}$ rays.

### 4.4 CONVERGENCE ANALYSIS.

In this section, we provide the theoretical analysis of InversionGNN, discussing its convergence properties in the discrete chemical space.

**Theorem 4.2** (Approximation Guarantee). *Under the assumptions in Sec. B, given an initial molecule $\boldsymbol{x}^0$ and a weight vector $\boldsymbol{\lambda}$, InversionGNN guarantees the following approximation when performing $T$ optimization rounds:*

$$\mathcal{L}^T \in \mathcal{A} := \left\{ \mathcal{L} \in \mathcal{O} \mid \mathcal{L} \preceq (\gamma \check{\lambda}^* + (1-\gamma)\check{\lambda}^0) \cdot \boldsymbol{\lambda}^{-1} \right\}, \tag{6}$$

*where $\gamma = \frac{1-\alpha^T}{(1-\alpha)N}$, $\boldsymbol{\lambda}^{-1} = (1/\lambda_i, \ldots, 1/\lambda_m)$, $\check{\lambda}^*$ and $\check{\lambda}^0$ is the maximum relative objective value $\check{\lambda}^t := \max \left\{ l_j^t \lambda_j \mid j \in [m] \right\}$ of $\boldsymbol{x}^*$ and $\boldsymbol{x}^0$.*

**Remark.** This theorem tells that InversionGNN can generate desired Pareto optimal molecules approximately within a few steps, and it is thus sample-efficient. Moreover, it implies that the Pareto optimal molecules is conditioned on the given weight vector, enabling chemical experts to design molecules that meet specific practical drug design requirements. Consequently, we can obtain an approximate set of Pareto optimal molecules that encompasses all possible trade-offs with various weight vectors. For more details about the proof, please refer to Appendix B.

**Time Complexity.** We did computational analysis in terms of inversion calls and computational complexity per iteration. (1) **Inversion Calls**. InversionGNN requires $O(TM)$ Oracle calls, where $T$ is the number of iterations. $M$ is the number of generated molecules, we have $M \leq NJ$, $N$ is the number of nodes in the scaffolding tree, for small molecule, N is very small. $J$ is the number of enumerated candidates in each node. (2) **Computational Complexity per Iteration.** The computation of $G^T G$ has runtime $O(m^2 n)$, where $n$ is the dimension of the gradients. With the current best QP solver (Zhang et al., 2021), we have a runtime of $O(m^3)$. Thus, the per-iteration time complexity is $O(m^2 n + m^3)$. Since in deep networks, usually $n \gg m$, InversionGNN does not significantly increase the computational cost in computing non-dominating gradient. The comparison of computational complexity between different methods is included in Appendix C.

## 5 EXPERIMENTS

In this section, we present our empirical findings which aim to answer the following questions:

**Q1**: *Can InversionGNN identify the Pareto optimal solution conditioned on desired weight?*

**Q2**: *Can InversionGNN explore the full Pareto front approximately?*

### 5.1 SYNTHETIC TASK

This section evaluates the InversionGNN via a commonly used synthetic objective in multi-objective optimization from Pareto MTL (Lin et al., 2019). *Different from the previous works in continuous space, we optimize the problem in discrete space.* We aim to minimize two non-convex objective

functions, denoted as:

$$l_1(\boldsymbol{x}) = 1 - e^{-\left\|\boldsymbol{x} - \frac{1}{\sqrt{n}}\right\|_2^2}, \quad l_2(\boldsymbol{x}) = 1 - e^{-\left\|\boldsymbol{x} + \frac{1}{\sqrt{n}}\right\|_2^2}, \tag{7}$$

where $\boldsymbol{x}$ represents a point in discrete Euclidean space, with its dimension set to $n = 20$. For these two objective functions, we are able to obtain the ground truth of the Pareto front.

**Metrics.** We use the standard metrics in multi-objective optimization: **Hypervolume (HV)** (Zitzler & Thiele, 1999) measures the volume in the objective space spanned by a set of non-dominated solutions and also represents Pareto diversity. We set the reference point as $(1, 1)$ in this task.

**Pareto Optimization Conditioned on Desired Weight(Q1).** To address this question, we adopt 5 weight vectors ($\lambda^{-1} Ray$). *The goal is to find the intersection points between the Pareto front and the weight $\lambda^{-1}$ ray.* We provide the details for generating weight vectors in Appendix D.3. For fair comparison, we incorporate Linear Scalarization into our framework as a baseline and term it as **I-LS**, refer to Appendix D.2. Due to the incompatibility of Genetic Algorithms and Bayesian Optimization with our framework, we adopted optimization techniques from two state-of-the-art drug discovery approaches,

Table 1: Hypervolume in Synthetic Task.

| Method | HV ($\uparrow$) |
|---|---|
| I-LS | $0.071_{\pm 0.003}$ |
| GAN-GA | $\underline{0.202}_{\pm 0.017}$ |
| HN-GFN | $0.187_{\pm 0.022}$ |
| InversionGNN | $\mathbf{0.328}_{\pm 0.001}$ |

GAN-GA (Abbasi et al., 2022) and MOGFN-AL (Jain et al., 2023), as baselines. InversionGNN and I-LS are optimized from random initialization for each weight vector. For GAN-GA, we evolve 300 iterations with population size of 40 and 10 offsprings. For MOGFN-AL, we start with 1000 points and running 100 optimization loops. We show the whole optimization process and Oracle calls in Figure 2. It illustrates that our InversionGNN framework not only captures the trade-offs among objective based on given weight vectors but also achieves the highest efficiency with 500 Oracle calls. In contrast, other baselines result in biased solutions.

**Exploring Full Pareto Front (Q2).** To address this question, we utilized 50 weight vectors to scan the entire Pareto front for InversionGNN and I-LS. As shown in Figure 3, our proposed InversionGNN can explore almost the full Pareto front. In contrast, the solutions of I-LS tend to cluster at the ends of the Pareto front, leaving certain regions unexplored. The results of GAN-GA and MOGFN-AL are shown in Figure 2, they merely find a small subset of the Pareto optimal solution. We also report HV and

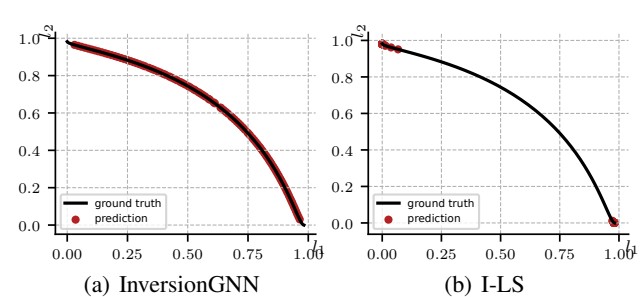

(a) InversionGNN      (b) I-LS

Figure 3: Pareto Front Search.

Oracle calls in Table 1, which shows that InversionGNN is able to cover the full Pareto front compared with baseline methods. GAN-GA and MOGFN-AL cost 12000 and 3000 Oracle calls respectively, but still achieve suboptimal performance.

## 5.2 MULTI-OBJECTIVE MOLECULAR OPTIMIZATION

In this section, we address the two questions by evaluating InversionGNN in discrete chemical space.

**Dataset.** We train the model on ZINC 250K dataset (Sterling & Irwin, 2015), which consists of 250K drug-like molecules extracted from the ZINC database. We select the substructures that appear more than 1000 times as the vocabulary set $S$, which consists of 82 frequent substructures.

**Implementation Details.** We implemented InversionGNN using Pytorch (Paszke et al., 2019). Both the size of substructure embedding and hidden size of GCN are $d = 100$. The depth of GNN $L$ is 3. In each generation, we keep $C = 10$ molecules for the next iteration. The learning rate is $1e - 3$ in training and inference procedure. We set the iteration $T$ to a large enough number and tracked the result. When Oracle calls budget is used up, we stop it. All results in the tables are from experiments up to $T = 50$ iterations.

Table 2: Weight Conditioned Molecular Optimization.

| Method | Nov($\uparrow$) | Div($\uparrow$) | APS($\uparrow$) | NU($\downarrow$) |
|---|---|---|---|---|
| MOEA/D | **100%** | n/a | $0.279_{\pm0.018}$ | $0.088_{\pm0.010}$ |
| NSGA-III | **100%** | n/a | $0.351_{\pm0.024}$ | $0.102_{\pm0.017}$ |
| MOGFN-PC | **100%** | $0.507_{\pm0.024}$ | $0.393_{\pm0.036}$ | $0.088_{\pm0.014}$ |
| HN-GFN | **100%** | $\mathbf{0.571}_{\pm0.032}$ | $0.418_{\pm0.022}$ | $0.072_{\pm0.015}$ |
| I-LS | **100%** | $0.541_{\pm0.007}$ | $0.529_{\pm0.006}$ | $0.049_{\pm0.002}$ |
| InversionGNN | **100%** | $0.435_{\pm0.009}$ | $\mathbf{0.648}_{\pm0.012}$ | $\mathbf{0.026}_{\pm0.001}$ |

Table 3: Multi-Objective Drug Discovery.

| Method | GSK3$\beta$ + JNK3 | | | | GSK3$\beta$+JNK3+QED+SA | | | |
|---|---|---|---|---|---|---|---|---|
| | Nov($\uparrow$) | Div($\uparrow$) | APS($\uparrow$) | Oracle($\downarrow$) | Nov($\uparrow$) | Div($\uparrow$) | APS($\uparrow$) | Oracle($\downarrow$) |
| LigGPT | **100%** | **0.845** | 0.271 | 100K+0 | **100%** | **0.902** | 0.378 | 100K+0 |
| GCPN | **100%** | 0.578 | 0.293 | 0+200K | **100%** | 0.596 | 0.450 | 0+200K |
| MolDQN | **100%** | 0.605 | 0.348 | 0+200K | **100%** | 0.597 | 0.365 | 0+200K |
| GA+D | **100%** | 0.657 | 0.608 | 0+50K | 97% | 0.681 | 0.632 | 0+50K |
| RationaleRL | **100%** | 0.700 | 0.795 | 25K+67K | 99% | 0.720 | 0.675 | 25K+67K |
| MARS | **100%** | 0.711 | 0.789 | 0+50K | **100%** | 0.714 | 0.662 | 0+50K |
| ChemBO | 98% | 0.702 | 0.747 | 0+50K | 99% | 0.701 | 0.648 | 0+50K |
| BOSS | 99% | 0.564 | 0.504 | 0+50K | 98% | 0.561 | 0.504 | 0+50K |
| LSTM | **100%** | 0.712 | 0.680 | 0+50K | **100%** | 0.706 | 0.672 | 0+50K |
| Graph-GA | **100%** | 0.634 | 0.825 | 0+25K | **100%** | 0.723 | 0.714 | 0+25K |
| DST | **100%** | 0.750 | 0.827 | 10K+5K | **100%** | 0.755 | 0.752 | 20K+5K |
| MOGFN-AL | **100%** | 0.673 | 0.742 | 50K+20K | **100%** | 0.711 | 0.621 | 50K+20K |
| RetMol | **100%** | 0.688 | 0.769 | 50K+5K | **100%** | 0.691 | 0.642 | 50K+5K |
| HN-GFN | **100%** | 0.784 | 0.725 | 50K+20K | **100%** | 0.733 | 0.638 | 50K+20K |
| I-LS | **100%** | 0.693 | 0.823 | 10K+5K | **100%** | 0.704 | 0.734 | 20K+5K |
| InversionGNN | **100%** | 0.768 | **0.841** | 10K+5K | **100%** | 0.769 | **0.773** | 20K+5K |

**Properties and Oracles.** (1) **QED** ranging in $[0, 1]$ that provides a quantitative assessment of a molecule's drug-likeness. (2) **SA** evaluates the ease of synthesizing a molecule, and is normalized to $[0, 1]$ (Gao & Coley, 2020). (3) **JNK3** is a member of the mitogen-activated protein kinase family, with scores ranging in $[0, 1]$. (4) **GSK3$\beta$** is an enzyme encoded by the GSK3$\beta$ gene in humans, and also has a range of $[0, 1]$. We utilize the RDKit package to evaluate QED and SA and evaluate GSK3$\beta$ and JNK3 following Li et al. (2018) and Jin et al. (2020b).

**Metrics.** We use standard metrics in molecular optimization. (1) **Novelty (Nov)** represents the proportion of generated molecules not in the training set. (2) **Top-K Diversity (Div)** (Bengio et al., 2021; Fu et al., 2022) of generated molecules is defined as the average pairwise Tanimoto distance between the Morgan fingerprints. (3) **Top-K Average Property Score (APS)** (Bengio et al., 2021; Fu et al., 2022) refers to the average score of the top-100 molecules. (4) **Oracle Calls** is represented as "$A + B$" which means allocate $A$ Oracle call budget for pretraining and $B$ for optimization.

**Molecular Optimization Conditioned on Desired Weight (Q1).** In this task, our goal is to evaluate that if InversionGNN can generate molecules conditioned on desired weight. Here we select two properties, GSK3$\beta$ and JNK3 and the 5 different weight vectors to serve as independent trials. **MOEA/D** (Zhang & Li, 2007) and **NSGA-III** (Deb & Jain, 2013) are two multi-objective Genetic Algorithms that also incorporate weight. We perform Genetic Algorithms over the latent space learned by JTVAE (Jin et al., 2018). In addition, we also report the **Non-Uniformity (NU)** that evaluates the distance between properties and weights vector. For MOEA/D and NSGA-III, we report

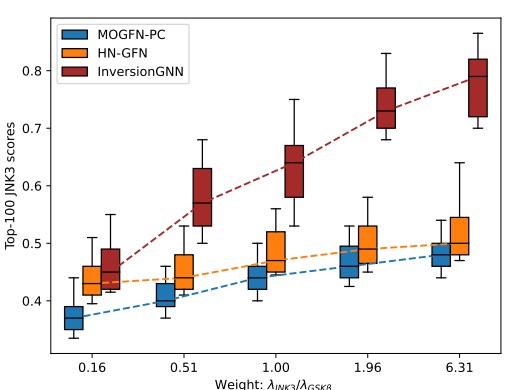

Figure 4: The distribution of Top-100 JNK3 scores.

the performance of the molecule with the lowest Non-Uniformity. **MOGFN-PC** (Jain et al., 2023) is a multi-objective molecular optimization method that scalarizes reward functions. **HN-GFN** (Zhu et al., 2024) is a multi-objective drug discovery method based on Bayesian optimization. For HN-GFN, I-LS, MOGFN-PC, and InversionGNN, we calculate the performance of the top-20 molecules in terms of Non-Uniformity per weight. For each weight, we compute the results separately and report the average results across all 5 trials. The results are shown in Table 2. InversionGNN outperforms most baselines by a significant margin. Our proposed InversionGNN achieves better uniformity and higher APS. We find that the diversity of InversionGNN is lower than that of HN-GFN and MOGFN-AL. A reasonable explanation is that the molecules produced by InversionGNN concentrate more around the desired weight with lower diversity. It is attributed to InversionGNN's ability to identify solutions that are more specifically related to the weight vector. We follow HN-GFN (Zhu et al., 2024) and visualize the top-100 JNK3 scores of the molecules generated by weight-based methods (InversionGNN, HN-GFN, and MOGFN-PC) conditioned on the 5 weight vectors in Figure 4. Even though all methods increase as the weight vector increases, InversionGNN outperforms the existing methods by a large margin.

**Multi-Objective Drug Discovery (Q2).** In this task, we use a set of weights to scan the Pareto front and evaluate the effectiveness of InversionGNN in synthesizing diverse molecules. We compare our InversionGNN with following baselines: (1) **LigGPT** (string-based distribution learning model) (Bagal et al., 2021); (2) **GCPN** (Graph Convolutional

Table 4: Hypervolume in Multi-Objective Drug Discovery.

| Method | HV ($\uparrow$) | |
|---|---|---|
| | GSK3$\beta$ + JNK3 | GSK3$\beta$+JNK3+QED+SA |
| MOGFN-AL | $0.567_{\pm0.057}$ | $\underline{0.377}_{\pm0.046}$ |
| HN-GFN | $\underline{0.592}_{\pm0.042}$ | $0.374_{\pm0.039}$ |
| DST | $0.497_{\pm0.019}$ | $0.353_{\pm0.024}$ |
| I-LS | $0.475_{\pm0.028}$ | $0.308_{\pm0.022}$ |
| InversionGNN | $\mathbf{0.763}_{\pm0.031}$ | $\mathbf{0.519}_{\pm0.038}$ |

Policy Network) (You et al., 2018); (3) **MolDQN** (Molecule Deep Q-Network) (Zhou et al., 2019); (4) **GA+D** (Genetic Algorithm with Discriminator network) (Nigam et al., 2020); (5) **MARS** (Markov Molecular Sampling) (Xie et al., 2021); (6) **RationaleRL** (Jin et al., 2020b); (7) **ChemBO** (Chemical Bayesian Optimization) (Korovina et al., 2020); (8) **BOSS** (Bayesian Optimization over String Space) (Moss et al., 2020); (9) **LSTM** (Long short term memory) (Brown et al., 2019); (10) **Graph-GA** (graph level genetic algorithm) (Brown et al., 2019); (11) **DST** (Differential Scaffolding Tree) (Fu et al., 2022);(12) **MOGFN-AL** (weight-conditional GFlowNets) (Jain et al., 2023); (13) **RetMol** (Retrieval-Based Generation) (Wang et al., 2023); (14) **HN-GFN** (Zhu et al., 2024). For I-LS and InversionGNN, we collect all the solutions of all weights and report the final results. The results are detailed in Table 3. InversionGNN exhibits superior performance compared to the majority of baselines. Diversity and APS is a common trade-off. Some methods encounter difficulties in simultaneously achieving high diversity scores and APS, due to their limited capacity to explore the chemical space. Despite LigGPT's achievement of high diversity, the notably low APS indicates its inability to effectively handle this trade-off. In contrast, InversionGNN shows superior performance on both metrics. In addition, we follow the recent multi-objective method HN-GFN (Zhu et al., 2024) and report the **Hypervolume (HV)** of the molecules generated by recent advanced MOMO approaches in Table 4, including HN-GFN, MOGFN-AL and DST. Higher HV indicates broader coverage of the objective space and higher Pareto diversity. The results show InversionGNN's ability to capture trade-offs among different properties.

### 5.3 ABLATION STUDY

**Optimization Process.** We demonstrate the optimization process of InversionGNN in Figure 5. We begin with an initial molecule $x_0$ and a weight vector $[1, 3]$. For clarity, we focus on two properties, JNK3 and GSK3$\beta$. At each step, we greedily add one substructure and display the corresponding molecular graphs, property scores, and the loss ratio: $ratio = \frac{l_{JNK3}}{l_{GSK3\beta}}$. As substructures are added, the property scores obtained by InversionGNN gradually increase and become more aligned with the weight vector. However, the I-LS significantly deviates from the weight vector. It demonstrates that InversionGNN is capable of finding Pareto optimal molecules conditioned on weight vector.

**Search Efficiency.** To understand the search efficiency of InversionGNN, we search the Pareto front with the weight vectors number of $(2, 5, 10, 15, 20)$. For GNK3$\beta$+JNK3, we allocate a $10K$ Oracle

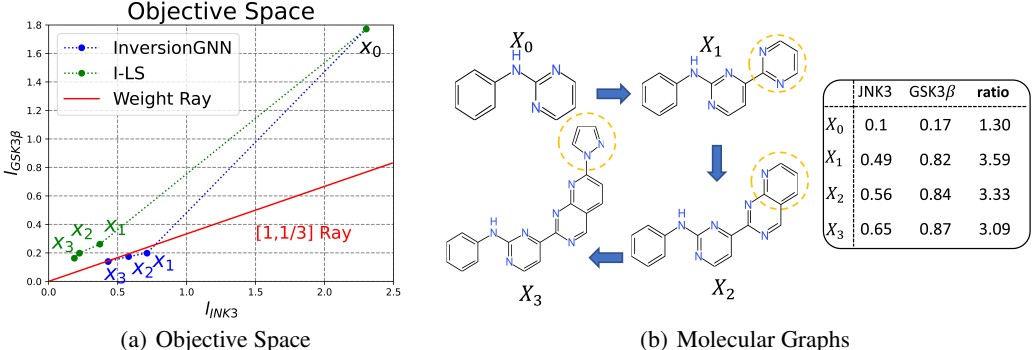

| (a) Objective Space | (b) Molecular Graphs |

Figure 5: Optimization process of InversionGNN on JNK3 and GSK3$\beta$ with the weight vector $[1, 3]$. (a) As substructures are added, the property scores obtained by InversionGNN increase and become more aligned with the weight vector. (b) Visualization of corresponding molecular graph.

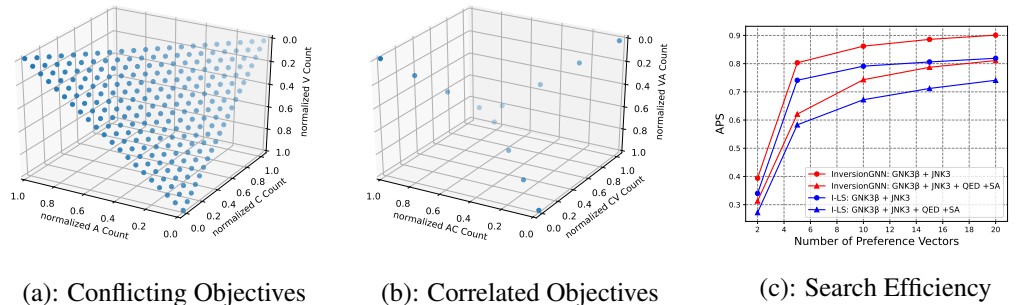

| (a): Conflicting Objectives | (b): Correlated Objectives | (c): Search Efficiency |

Figure 6: Pareto front obtained by Inversion GNN on (a) Conflicting Objectives, and (b) Correlated Objectives. (c) Search Efficiency in Multi-Objective Drug Discovery. The number of weights represents the search scope, and the number of Oracle calls grows with the number of weights.

call budget for surrogate Oracle pretraining, and $N_{weight} \times 1K$ Oracle call budget for optimization. For the optimization involving GNK3$\beta$+JNK3+QED+SA, the pretraining budget is fixed at $20K$, and $N_{weight} \times 1K$ Oracle call budget for optimization. As illustrated in Figure 6 (c), APS increases with growing weight vectors in multi-objective molecular optimization task.

**Conflicting and Correlated Objectives.** For fair comparison, we follow the synthetic sequence design task in MOGFN (Jain et al., 2023). The task consists of generating strings with the objectives given by occurrences of a set of $d$ n-grams. We consider a vocabulary of size 4, with 3 characters ['C', 'V',' A'] and a special token to indicate the end of the sequence. The objectives are defined by the number of occurrences of a given set of n-grams in the sequence. Therefore, for conflicting objectives setting, we use 3 the Unigrams task ['C',' V',' A']. InversionGNN adequately models the trade-off between conflicting objectives as illustrated by the generated Pareto front in Figure 6 (a). For the 3 Bigrams task with correlated objectives ['CV',' VA', 'AC'], Figure 6 (b) demonstrates InversionGNN can simultaneously maximize multiple correlated objectives.

## 6  CONCLUSION

In this work, we propose a novel dual-path InversionGNN for multi-objective molecular optimization, which is effective and sample-efficient. In the direct prediction path, we incorporate complicated chemical knowledge. In the inversion generation path, we decode the acquired knowledge with the gradient w.r.t the molecule structure. To tackle the complicated multi-property chemical knowledge, we relax the discrete molecule optimization into a locally differentiable Pareto optimization problem. Through extensive experimental evaluations, we have demonstrated the effectiveness and sample efficiency of InversionGNN in multi-objective drug discovery.

ACKNOWLEDGMENTS

This work was supported by HKUST – HKUST(GZ) Cross-campus Collaborative Research Scheme under the Guangdong "1+1+1" Joint Funding Program.

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

# A    PARETO OPTIMAL

In this work, we consider $m$ tasks described by $f(\boldsymbol{x}) := [f_i(\boldsymbol{x})] : \mathbb{R}^n \to \mathbb{R}^m$ for any point $\boldsymbol{x}$ in *Solution Space* $\mathbb{R}^n$, where each $f_i(\boldsymbol{x}) : \mathbb{R}^n \to \mathbb{R}, i \in [m]$ represents the performance of the $i$-th task to be maximized. Given an desired target $\boldsymbol{y} \in \mathbb{R}^m$, we set a non-negative objective function $\mathcal{L}(f(\boldsymbol{x}), \boldsymbol{y}) = [l_1, \ldots, l_m]^\mathsf{T} : \mathbb{R}^m \to \mathbb{O}^m$ to be a non-negative objective function mapping the *Value Space* $\mathbb{R}^m$ to the *Objective Space* $\mathbb{O}^m$, where $l_i$ for $i \in [m]$ is the objective function of the $i$-th task. Hence, maximizing the performance $f(\boldsymbol{x})$ is equivalent to minimizing the objective function. Consequently, we have $l_i^{\boldsymbol{x}'} - l_i^{\boldsymbol{x}} \geq 0$ if $f_i(\boldsymbol{x}') \leq f_i(\boldsymbol{x})$ for two points $\boldsymbol{x}, \boldsymbol{x}' \in \mathbb{R}^n$.

For any two points $\boldsymbol{x}, \boldsymbol{x}' \in \mathbb{R}^n$, $\boldsymbol{x}$ dominates $\boldsymbol{x}'$, denoted by $\mathcal{L}^{\boldsymbol{x}'} \succeq \mathcal{L}^{\boldsymbol{x}}$, if and only if $\mathcal{L}^{\boldsymbol{x}'} - \mathcal{L}^{\boldsymbol{x}} \in \mathbb{R}_+^m$, where $\mathbb{R}_+^m := \{\mathcal{L} \in \mathbb{O}^m | l_i \geq 0, \forall i \in [m]\}$. The partial ordering $\mathcal{L}^{\boldsymbol{x}'} \succeq \mathcal{L}^{\boldsymbol{x}}$ implies $l_i^{\boldsymbol{x}'} - l_i^{\boldsymbol{x}} \geq 0, \forall i \in [m]$. When $\boldsymbol{x}$ strictly dominates $\boldsymbol{x}'$, denoted by $\mathcal{L}^{\boldsymbol{x}'} \succ \mathcal{L}^{\boldsymbol{x}}$, it means there is at least one $i$ for which $l_i^{\boldsymbol{x}'} - l_i^{\boldsymbol{x}} > 0$. Geometrically, $\mathcal{L}^{\boldsymbol{x}'} \succ \mathcal{L}^{\boldsymbol{x}}$ means that $\mathcal{L}^{\boldsymbol{x}'}$ lies in the positive cone pivoted at $\mathcal{L}^{\boldsymbol{x}}$, i.e. $\mathcal{L}^{\boldsymbol{x}'} \in \{\mathcal{L}^{\boldsymbol{x}}\} + \mathbb{R}_+^m := \{\mathcal{L}^{\boldsymbol{x}} + \mathcal{L} \mid \mathcal{L} \in \mathbb{R}_+^m\}$. A point $\boldsymbol{x}^*$ is said to be **Pareto optimal** if $\boldsymbol{x}^*$ is not dominated by any other points in $\mathbb{R}^n$. Similarly, $\boldsymbol{x}^*$ is locally Pareto optimal if $\boldsymbol{x}^*$ is not dominated by any other points in the neighborhood of $\boldsymbol{x}^*$, i.e. $\mathcal{N}(\boldsymbol{x}^*)$. The set of all Pareto optimal solutions is defined as:

$$\mathcal{P} := \left\{\boldsymbol{x}^* \in \mathbb{R}^n \mid \forall \boldsymbol{x} \in \mathbb{R}^n - \{\boldsymbol{x}^*\}, \mathcal{L}^{\boldsymbol{x}^*} \not\succeq \mathcal{L}^{\boldsymbol{x}}\right\}, \tag{8}$$

where $\mathcal{L}^{\boldsymbol{x}^*} \not\succeq \mathcal{L}^{\boldsymbol{x}}$ represents $\boldsymbol{x}^*$ is not dominated by other point $\boldsymbol{x}$. The set of multi-objective values of the Pareto optimal solutions is called **Pareto front**:

$$\mathcal{F} := \left\{\mathcal{L}^{\boldsymbol{x}^*} \mid \boldsymbol{x}^* \in \mathcal{P}\right\}. \tag{9}$$

In Multi-Objective Optimization, the ideal goal is to identify a set of Pareto solutions that cover all the possible preferences among objectives.

# B    THEORETICAL ANALYSIS

In this section, we discuss the theoretical properties of InversionGNN Algorithm.

## B.1    ASSUMPTIONS AND KEY LEMMAS

**Assumption B.1** (Molecule Size Bound). The sizes (i.e., number of substructures) of all the scaffolding trees generated are upper bounded by $N$.

We focus on small molecule optimization; the target molecular properties would decrease significantly when the molecule size is too large (Bickerton et al., 2012), e.g., QED. To perform a convergence analysis, we initially establish several assumptions to characterize the geometry of the objective landscape.

**Definition B.2** (Non-Uniformity). For any point $\boldsymbol{x} \in \mathbb{R}^n$, the Non-Uniformity of its objective values $\mathcal{L}$ in relation to a given weight vector $r \in \mathbb{R}^m$ as:

$$\mu_r(\mathcal{L}) = \sum_{i=1}^m \hat{l}_i \log\left(\frac{\hat{l}_i}{1/m}\right) = \mathrm{KL}\left(\hat{\mathcal{L}} \mid \frac{\mathbf{1}}{m}\right), \tag{10}$$

where $\hat{\mathcal{L}} = [\hat{l}_1, \ldots, \hat{l}_m]^\mathsf{T}$ and $\hat{l}_i$ is the weighted normalization $\hat{l}_i = \frac{r_i l_i}{\sum_{i'=1}^m r_{i'} l_{i'}}$.

The Kullback-Leibler (KL) divergence of $\hat{\mathcal{L}}$ from the uniform distribution $\dfrac{\mathbf{1}}{m}$ characterizes non-uniformity. When the objective value fulfills the weight condition, we have $\mu_r(\mathcal{L}) = 0$; otherwise, $\mu_r(\mathcal{L}) > 0$. Consequently, we prefer a lower $\mu_r(\mathcal{L})$.

**Definition B.3** (Dominant Set). Given $\boldsymbol{x}^t$ in chemical space $\mathcal{X}$ at the $t$-th iteration, we define a Dominant Set $\mathcal{V}_{\preceq \mathcal{L}^t} \subset \mathbb{R}^m$ that contains all attainable multi-objective values that dominate the $\mathcal{L}^t$ as:

$$\mathcal{V}_{\preceq \mathcal{L}^t} = \left\{\mathcal{L} \in \mathcal{O} \mid \mathcal{L} \preceq \mathcal{L}^t\right\}. \tag{11}$$

**Definition B.4** (Uniform Set). Given a molecule $x^t$ in the chemical space $\mathcal{X}$ at the $t$-th iteration and a specified weight vector $r \in \mathbb{R}^m$, we define a Uniform Set $\mathcal{M}_{\mathcal{L}^t}^r \subset \mathbb{R}^m$ that contains all attainable multi-objective values demonstrating enhanced uniformity compared to $\mathcal{L}^t$ as:

$$\mathcal{M}_{\mathcal{L}^t}^r = \left\{ \mu_r(\mathcal{L}) \leq \mu_r(\mathcal{L}^t) \right\}. \tag{12}$$

**Definition B.5** (Admissible Set). Given a molecule $x^t$ in the chemical space $\mathcal{X}$ at the $t$-th iteration and a specified weight vector $r \in \mathbb{R}^m$, we define a bounded Admissible Set $\mathcal{A}_{\mathcal{L}^t}^r \subset \mathbb{R}^m$ as:

$$\mathcal{A}_{\mathcal{L}^t}^r = \left\{ \mathcal{L} \in \mathcal{O} \mid \mathcal{L} \preceq \check{\mathcal{L}}^t \right\}, \tag{13}$$

where $\check{\mathcal{L}}^t = \check{r}^t (1/r_1, \cdots, 1/r_m)$, and $\check{r}^t = \max \left\{ \mathcal{L}_j^t r_j \mid j \in [m] \right\}$.

Clearly, the admissible set contains all the points in $\mathcal{O}$ that dominate the $\mathcal{L}^t$, i.e. $\mathcal{V}_{\preceq \mathcal{L}^t} \subset \mathcal{A}_{\mathcal{L}^t}^r$. Moreover, when $\mu_r(\mathcal{L}) > 0$, it also contains points exhibiting superior uniformity compared to $\check{\mathcal{L}}^t$, i.e. $\mathcal{A}_{\mathcal{L}^t}^r \cap \mathcal{M}_{\mathcal{L}^t}^r \neq \emptyset$. Consequently, the admissible set encompasses the desired solution for the subsequent iteration, fulfilling both uniformity and dominating properties.

As illustrated by Mahapatra & Rajan (2020), a descent direction $d_{nd}$ will be oriented towards the weight-specific Pareto front and within a confined admissible set. Since we perform $K$ descents in each iteration, we restructure properties for InversionGNN in the molecular optimization problem as follows:

**Lemma B.6** (Bounded Objective Space for the Next Iteration). *There exists a step size $\eta_0 > 0$, such that for every $\eta \in [0, \eta_0]$, InversionGNN employs $\widetilde{\mathcal{T}}_{x^t} = \widetilde{\mathcal{T}}_{x^t} - \eta d_{nd}$ to update differentiable scaffolding tree until convergence. Subsequently, we greedily sample a molecule as $x^{t+1}$ from $\widetilde{\mathcal{T}}_{x^t}^K$ by adding a substructure, if the solution is nonempty. The multi-objective value $\mathcal{L}^{t+1}$ of the new solution point $x^{t+1}$ lies in the $t$-th admissible set:*

$$\mathcal{L}^{t+1} \in \mathcal{A}_{\mathcal{L}^t}^r. \tag{14}$$

Following Theorem 2 in EPO (Mahapatra & Rajan, 2020), the empirical loss $\mathcal{L}_{\widetilde{\mathcal{T}}}$ of scaffolding tree $\widetilde{\mathcal{T}}_{x^t}^K$ lies in the t-th admissible set $\mathcal{A}_{\mathcal{L}^t}^r$. Since we greedily sample a molecule as $x^{t+1}$ from $\widetilde{\mathcal{T}}_{x^t}^K$, thus we have $\mathcal{L}^{t+1} \preceq \mathcal{L}_{\widetilde{\mathcal{T}}}$. Therefore, it follows that $\mathcal{L}^{t+1} \in \mathcal{A}_{\mathcal{L}^t}^r$. It demonstrates that for a molecule $x^t$ at the $t$-th iteration, InversionGNN selects a molecule as $x^{t+1}$ from $x^t$'s neighborhood set $\mathcal{N}(x^t)$, moving towards improved uniformity and dominating properties. It provides a theoretical guarantee for the quality of the solution.

**Corollary B.7** (Convergence of Admissible Set). *The sequence of relative maximum values $\check{r}^t$ obtained by descending against the adjusted gradient $d_{nd}$ is monotonic with $\check{r}^{t+1} \leq \check{r}^t$, which means*

$$\mathcal{A}_{\mathcal{L}^t}^r \subset \mathcal{A}_{\mathcal{L}^{t+1}}^r, \tag{15}$$

*and the sequence of bounded sets $\{\mathcal{A}_{\mathcal{L}^{t+1}}^r\}$ converges.*

Since $\mathcal{L}^{t+1} \in \mathcal{A}_{\mathcal{L}^t}^r$, we naturally get $\check{r}^{t+1} \leq \check{r}^t$, thus we have $\mathcal{A}_{\mathcal{L}^t}^r \subset \mathcal{A}_{\mathcal{L}^{t+1}}^r$. It demonstrates the monotonicity of $\check{r}^t$. Suppose InversionGNN selects a molecule as $x^{t+1}$ from $x^t$'s neighborhood set $\mathcal{N}(x^t)$, where the lowest $\check{r}^{t+1}$ is precisely determined, i.e., finding a solution that maximizes $\left| \check{r}^{t+1} - \check{r}^t \right|$.

**Assumption B.8** (Submodularity and Smoothness). Suppose $x_{t-1}, x_t, x_{t+1}$ are generated successively by InversionGNN via growing a substructure on the scaffolding tree. We assume that the corresponding objective gain (i.e., $\triangle \check{r}^t$) satisfies the diminishing returns property:

$$\check{r}^{t-1} - \check{r}^t \geq \check{r}^t - \check{r}^{t+1}, \quad (submodularity) \tag{16}$$

Submodularity plays the role of concavity/convexity in the discrete regime. On the other hand, we specify the curvature ratio of the objective function $\mathcal{L}$ by assuming

$$\check{r}^t - \check{r}^{t+1} \geq \alpha(\check{r}^{t-1} - \check{r}^t), \quad 0 < \alpha < 1 - \frac{1}{N} < 1. \quad (curvature) \tag{17}$$

The choice of submodularity as an assumption for our analysis was motivated by experimental observations. For instance, in the optimization process, we noticed that the objective values, such as QED, can rapidly increase to a high point with just a few iterations. However, as we added more atoms with InversionGNN, the growth rate began to decrease, i.e. the return is diminishing when the property scores are reaching the upper bound. This trend is observed in many properties and provides insight into our assumption.

## B.2 THEOREM

**Theorem B.9** (Approximation Guarantee). *Under the assumptions stated in Sec. B, Given an initial molecule $\boldsymbol{x}^0$ and weight vector $\boldsymbol{r}$, InversionGNN guarantees the following approximation when performing $T$ optimization rounds:*

$$\mathcal{L}^T \in \mathcal{A} := \left\{ \mathcal{L} \in \mathcal{O} \mid \mathcal{L} \preceq (\gamma \check{r}^* + (1-\gamma)\check{r}^0) \cdot \boldsymbol{r}^{-1} \right\}, \tag{18}$$

*where $\gamma = \frac{1-\alpha^T}{(1-\alpha)N}$, $\boldsymbol{r}^{-1}$ is $(1/r_i, \ldots, 1/r_m)$, $\check{r}^*$ and $\check{r}^0$ is the maximum relative objective value $\check{r}^t := \max \left\{ l_j^t r_j \mid j \in [m] \right\}$ of $\boldsymbol{x}^*$ and $\boldsymbol{x}^0$.*

*Proof.* In the following steps of the proof, to simplify mathematical notation, we substitute $r^t$ for $\check{r}^t$. Starting from $\boldsymbol{x}^0$, suppose the path to optimum $\boldsymbol{x}^*$ with the weight $r$ is

$$\boldsymbol{x}^0 \to \boldsymbol{x}^1 \to \boldsymbol{x}^2 \to \cdots \to \boldsymbol{x}^k = \boldsymbol{x}^*, \tag{19}$$

where each step, one substructure is added.

For InversionGNN, we run $T \in [k, N]$ iterations, and the path produced by InversionGNN is

$$\hat{\boldsymbol{x}}^0(\boldsymbol{x}^0) \to \hat{\boldsymbol{x}}^1 \to \hat{\boldsymbol{x}}^2 \to \cdots \to \hat{\boldsymbol{x}}^T, \quad \text{where } T \geq k. \tag{20}$$

For the optimum $\boldsymbol{x}^*$, based on the submodularity in Assumption B.8 we have

$$k \left( r^0 - r^1 \right) \geq \sum_{j=1}^k (r^{j-1} - r^j) = r^0 - r^k = r^0 - r^*. \tag{21}$$

From assumption B.1, it follows that

$$r^0 - r^1 \geq \frac{1}{k}(r^0 - r^*) \geq \frac{1}{N}(r^0 - r^*). \tag{22}$$

For the molecular $\boldsymbol{z}^T$ found by InversionGNN, based on curvature ratio in Assumption B.8 we have

$$\hat{r}^{T-1} - \hat{r}^T \geq \alpha \left( \hat{r}^{T-2} - \hat{r}^{T-1} \right) \geq \cdots \geq \alpha^{T-1} \left( \hat{r}^0 - \hat{r}^1 \right). \tag{23}$$

Then we have

$$\hat{r}^0 - \hat{r}^T = \sum_{j=1}^T (r^{j-1} - r^j) \geq \sum_{j=1}^T \alpha^{j-1}(\hat{r}^0 - \hat{r}^1) = \frac{1-\alpha^T}{1-\alpha} \left( (\hat{r}^0 - \hat{r}^1) \right). \tag{24}$$

Since InversionGNN pick up a molecule as $\boldsymbol{x}^{t+1}$ from $\boldsymbol{x}^t$'s neighborhood set $\mathcal{N}(\boldsymbol{x}^t)$ with lowest $\check{r}^{t+1}$ is exactly solved, i.e. $\hat{r}^1 \leq r^1$, and $\hat{r}^0 = r^0$. Thus we have

$$\hat{r}^0 - \hat{r}^1 \geq r^0 - r^1. \tag{25}$$

From Eq. 22, Eq. 24 and Eq. 25, it follows that

$$r^0 - \hat{r}^T \geq \frac{1-\alpha^T}{(1-\alpha)N}(r^0 - r^*). \tag{26}$$

Thus we have:

$$\hat{r}^T \leq \frac{1-\alpha^T}{(1-\alpha)N}r^* + (1 - \frac{1-\alpha^T}{(1-\alpha)N})r^0. \tag{27}$$

Finally, it follows that

$$\mathcal{L}^T \in \mathcal{A} := \left\{ \mathcal{L} \in \mathcal{O} \mid \mathcal{L} \preceq (\gamma r^* + (1-\gamma)r^0) \cdot \boldsymbol{r}^{-1} \right\}, \tag{28}$$

$\square$

## C  Theoretical comparison of computational complexity

The current multi-objective molecular optimization methods can be categorized into three classes: (1) gradient-based Pareto search (InversionGNN), (2) multi-objective genetic algorithm, and (3) multi-objective Bayesian optimization, as illustrated in Section 1. We provide a theoretical analysis of their computational bottlenecks during one iteration:

- Gradient-based Pareto search: If we perform $K$ gradient descent steps in each iteration, the time complexity is $O(K(m^2n + m^3))$, where $n$ is the dimension of the gradients and $m$ is the number of objectives, usually $n \gg m$.
- Genetic Algorithm: The computational bottleneck lies in the non-dominated sorting, which has a time complexity of $O(mp^2)$, where $p$ is the population size and $m$ is the number of objectives.
- Bayesian Optimization: The computational bottleneck lies in training the Gaussian process, which has a time complexity of $O(q^3)$, where $q$ is the number of training data points.

## D  Implementations Details

### D.1  Molecular Representation

A molecular graph is a representation of a molecule, consisting of atoms as nodes and chemical bonds as edges. Nonetheless, challenges such as chemical validity constraints, ring integrity, and extensive calculations hinder the explicit reconstruction of potential connectivity. To tackle this, Jin et al. (2018) introduced a scaffolding tree, a spanning tree that employs nodes as *substructures* to model a higher-level representation of a molecule.

A scaffolding tree, denoted by $\mathcal{T}_{\boldsymbol{x}} = \{\mathbf{N}, \mathbf{A}, \mathbf{w}\}$, serves as a high-level representation of a molecule $\boldsymbol{x} \in \mathcal{X}$. Each node is a member of the substructure set $\mathcal{S}$ (also referred to as the vocabulary set). $\mathcal{T}_{\boldsymbol{x}}$ consists of three components: (i) the node indicator matrix defined as $\mathbf{N} \in \{0,1\}^{K \times |S|}$, where each row of $N$ is a one-hot vector indicating the substructure to which the node belongs; (ii) the adjacency matrix denoted by $\mathbf{A} \in \{0,1\}^{K \times K}$, where $\mathbf{A}_{ij} = 1$ when the $i$-th and the $j$-th nodes are connected, and 0 when they are unconnected; (iii) $\mathbf{w} = [1, \ldots, 1]^\top \in \mathbb{R}^K$, signifies that the $K$ nodes are equally weighted. We convert molecules to scaffolding tree for training the surrogate Oracle.

To modify the scaffolding tree $\mathcal{T}_{\boldsymbol{x}}$, we employ its differentiable version, $\widetilde{\mathcal{T}}_{\boldsymbol{x}}$, as proposed by Fu et al. (2022). The basic scaffolding tree, $\mathcal{T}_{\boldsymbol{x}}$, can be transformed into a tree containing $K + K_{expand}$ nodes, denoted by $\widetilde{\mathcal{T}}_{\boldsymbol{x}} = \{\widetilde{\mathbf{N}}, \widetilde{\mathbf{A}}, \widetilde{\mathbf{w}}\}$, through the addition of an expansion node set $\mathcal{V}_{expand} = \{u_v \mid v \in \mathcal{V}_{\mathcal{T}_{\boldsymbol{x}}}\}$, where $|\mathcal{V}_{expand}| = K_{expand} = K$. It is crucial to note that $\widetilde{\mathcal{T}}_{\boldsymbol{x}}$ contains learnable parameters, which can be interpreted as conditional probability. It can be utilized to sample a new tree through processes such as node shrinking, replacement, or expansion. Each scaffolding tree corresponds to multiple molecules, as substructures can be combined in various ways.

**Assemble the Scaffolding Tree into Molecule.**  We conduct following steps: (a) Ring-atom connection. When connecting atom and ring in a molecule, an atom can be connected to any possible atoms in the ring. Ring-ring connection. (b) When connecting ring and ring, there are two general ways, (1) one is to use a bond (single, double, or triple) to connect the atoms in the two rings. (2) another is two rings share two atoms and one bond.

### D.2  Details of I-LS

The baseline method with Linear Scalarization (LS) is summaried in Algorithm 2. LS dose not compute the non-dominated gradient $d_{nd}$ but instead linearly weights the gradient using weights, i.e., $d = Gr$. We keep the remaining parts consistent with InversionGNN.

### D.3  Weight Vector

In this section, we describe the detailed process for generating a weight vector uniformly distributed in the objective space. We list the algorithm 3 and 4 for 2 and 4 objectives.

---

**Algorithm 2:** Linear Scalarization (I-LS)

---

**Input:** Input molecule $\boldsymbol{x}^0 \in \mathcal{X}$, weight vector $r \in \mathbb{R}^m$, and step size $\eta > 0$.
**Output:** Generated Molecule $\boldsymbol{x}^*$.

1  Initialization.
2  Train surrogate Oracle according to Eq. 3.
3  **for** $t = 0, \ldots, T$ **do**
4       Convert molecule $\boldsymbol{x}^t$ to scaffolding tree $\widetilde{\mathcal{T}}^0_{\boldsymbol{x}^t}$;
5       **for** $k = 0, \ldots, K$ **do**
6           Compute gradients of each property objective w.r.t. $\widetilde{\mathcal{T}}^k_{\boldsymbol{x}^t}$: $G = \nabla \mathcal{L} = [g_1, \ldots, g_m]$;
7           Calculate the direction of descent $d_{ls} = Gr$;
8           Update the differentiable scaffolding tree using $\widetilde{\mathcal{T}}^{k+1}_{\boldsymbol{x}^t} = \widetilde{\mathcal{T}}^k_{\boldsymbol{x}^t} - \eta d_{ls}$.
9       **end**
10      Sample discrete $\mathcal{T}_{\boldsymbol{x}^{t+1}}$ from continuous $\widetilde{\mathcal{T}}^K_{\boldsymbol{x}^t}$ and assemble it to molecule $\boldsymbol{x}^{t+1}$;
11 **end**

---

**Algorithm 3:** Generate Weight for Two Objective

---

1  Generate a uniformly distributed variable, $u$, ranging from 0 to 1.
2  Compute coordinates' angle: $\theta = \frac{\pi}{2} u$.
3  Compute Cartesian coordinates: $r_1 = \cos\theta, r_2 = \sin\theta$.

---

### D.4  INVERSIONGNN SETUP

Most of the settings follow the DST (Fu et al., 2022). We implemented InversionGNN using Pytorch 1.7.1, Python 3.7.9. Both the size of substructure embedding and hidden size of GCN are $d = 100$. The depth of GNN $L$ is 3. In each generation, we keep $C = 10$ molecules for the next iteration. The learning rate is 1e-3 in training and inference procedure. We set the iteration $T$ to a large enough number and tracked the result. When Oracle calls budget is used up, we stop it. All results in the tables are from experiments up to $T = 50$ iterations. For I-LS, We only replace the objective function in InversionGNN with Linear Scalarization, and other settings are consistent with InversionGNN.

### D.5  BASELINES

In this section, we describe the detailed experimental setting for baseline methods. Most of the settings follow the original papers.

- **LigGPT** is a string-based distribution learning model with a Transformer as decoder (Bagal et al., 2021), we trained it for 10 epochs using the Adam optimizer with a learning rate of $6e - 4$.

- **GCPN** (Graph Convolutional Policy Network) (You et al., 2018) leveraged graph convolutional network and policy gradient to optimize the reward function that incorporates target molecular properties and adversarial loss. we trained it using Adam optimizer with 1e-3 initial learning rate, and batch size is 32.

- **MolDQN** (Molecule Deep Q-Networks) (Zhou et al., 2019) formulate the molecule generation procedure as a Markov Decision Process (MDP) and use Deep Q-Network to solve it. Adam is trained Adam optimizer with 1e-4 as the initial learning rate, $\epsilon$ is annealed from 1 to 0.01 in a piecewise linear way.

- **GA+D** (Genetic Algorithm with Discriminator network) (Nigam et al., 2020) uses a deep neural network as a discriminator to enhance exploration in a genetic algorithm and is trained using the Adam optimizer with a learning rate of $1e - 3$, $\beta$ is set it to 10.

- **MARS** (Xie et al., 2021) leverage Markov chain Monte Carlo sampling (MCMC) on molecules with an annealing scheme and an adaptive proposal. It is trained using Adam optimizer with 3e-4 initial learning rate.

---

**Algorithm 4:** Generate Weight for Four Objective

---

1 Generate two uniformly distributed variables, $u$, $v$ and $z$, ranging from 0 to 1.
2 Compute spherical coordinates' inclination angle and azimuth angle: $\theta = \frac{\pi}{2}u$, $\phi = \arccos v$, $\sigma = \arccos z$.
3 Compute Cartesian coordinates: $r_1 = \sin\phi\cos\theta\sin\sigma$, $r_2 = \sin\phi\sin\theta\sin\sigma$, $r_3 = \sin\sigma\cos\phi$, $r_4 = \cos\sigma$.

---

- **RationaleRL** (Jin et al., 2020b) is a deep generative model that grows a molecule atom-by-atom from an initial rationale (subgraph). It is trained using Adam optimizer on both pre-training and fine-tuning with initial learning rates of 1e-3, 5e-4, respectively. The annealing rate is 0.9.

- **ChemBO** (chemical Bayesian optimization) (Korovina et al., 2020) leverage Bayesian optimization. It also explores the synthesis graph in a sample-efficient way and produces synthesizable candidates. Following the default setting in the original paper, the number of steps of acquisition optimization is set to 20. The initial pool size is set to 20, while the maximal pool size is 1000.

- **BOSS** (Bayesian Optimization over String Space) (Moss et al., 2020) builds a Gaussian process surrogate model based on Sub-sequence String Kernel, which naturally supports SMILES strings with variable length, and maximizing acquisition function efficiently for spaces with syntactical constraints. The population size is set to 100, the generation (evolution) number is set to 100.

- **DST** (Differentiable Scaffolding Tree) (Fu et al., 2022) utilizes a learned knowledge network to convert discrete chemical structures to locally differentiable ones. DST enables a gradient-based optimization on a chemical graph structure by back-propagating.

- **MOGFN-PC** (weight-conditional GFlowNets) (Jain et al., 2023) is a Reward-conditional GFlowNets based on Linear Scalarization. They introduce the Weighted-log-sum that can help in scenarios where all objectives are to be optimized simultaneously, and the scalar reward from Weighted-Sum can be dominated by a single reward.

- **HN-GFN** (Zhu et al., 2024) is a multi-objective drug discovery method based on Bayesian optimization. It leverages the hypernetwork-based GFlowNets as an acquisition function optimizer.

- **RetMol** (Retrieval-Based Molecular Generation) (Wang et al., 2023) retrieves and fuses the exemplar molecules with the input molecule, which is trained by a new selfsupervised objective that predicts the nearest neighbor of the input molecule.

## E  ADDITIONAL EXPERIMENTAL RESULTS

### E.1  MOLECULES GENERATED BY INVERSIONGNN

We provide several molecules synthesized via the InversionGNN approach.

(1) **Molecules with JNK3 and GSK3$\beta$ scores**. Each score independently represents the respective values for JNK3 and GSK3$\beta$, see Figure 8.

(2) **Molecules with highest average QED, normalized-SA, JNK3 and GSK3$\beta$ scores**. These four scores symbolize the values for QED, normalized SA, JNK3, and GSK3$\beta$, respectively, see Figure 9.

### E.2  MOLECULES GENERATED BY LINEAR SCALARIZATION

We exhibit the molecular graphs produced through Linear Scalarization. Alongside these visual representations, their corresponding property scores are included, and the loss ratio is calculated using the formula $ratio = \frac{l_{JNK3}}{l_{GSK3\beta}}$. This supplementary information further elaborates on the experimental results outlined in Section 6.3 of the paper, as illustrated in Figure 10.

### E.3  MOLECULES GENERATED BY INVERSIONGNN

The Oracle requires realistic optimization tasks, which can often be time-consuming. To further verify the Oracle efficiency, we explore a special setting of molecule optimization where the budget

of Oracle calls is limited to a fixed number (2K, 5K, 10K, 20K, 50K) and compare the optimization performance. For GCPN, MolDQN, GA+D, and MARS, the number of learning iterations is determined by the Oracle call budget. To ensure a fair comparison with DST, InversionGNN, and I-LS utilize approximately 80% of the budget to label the dataset (i.e., for training the GNN), reserving the remaining budget for de novo design. Specifically, for each budget (2K, 5K, 10K, 20K, and 50K), we allocate 1.5K, 4K, 8K, 16K, and 40K Oracle calls, respectively, for labeling the data used in GNN training. Figure 7 illustrates the APS of the top 100 molecules across different Oracle budgets. Notably, InversionGNN shows a significant advantage compared to all the baseline methods in limited-budget settings.

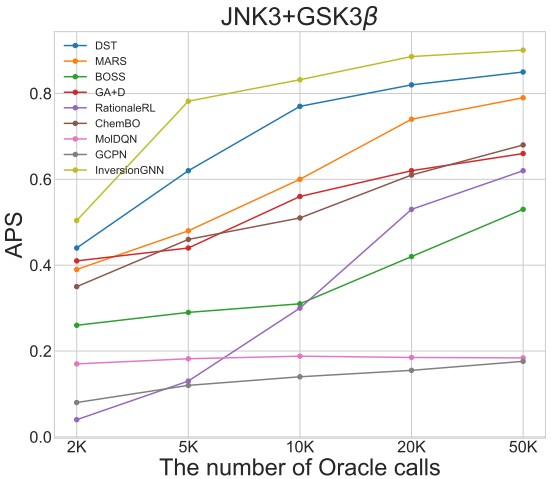

Figure 7: Oracle Efficiency Test.

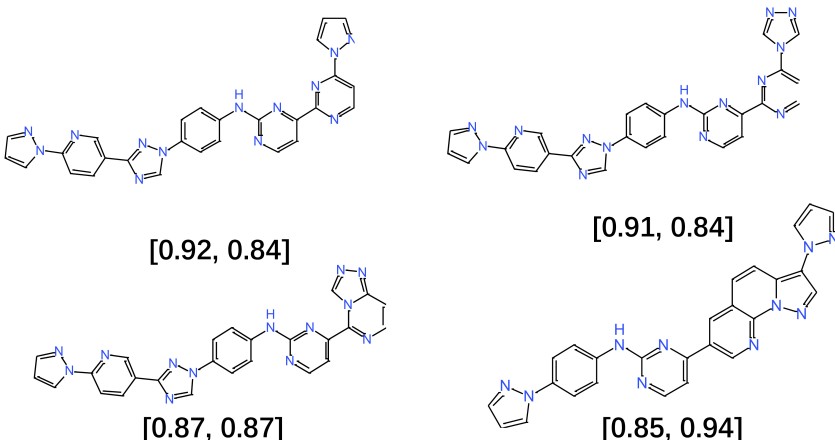

Figure 8: Generated molecules by InversionGNN. These four scores symbolize the values for JNK3 and GSK3$\beta$, respectively.

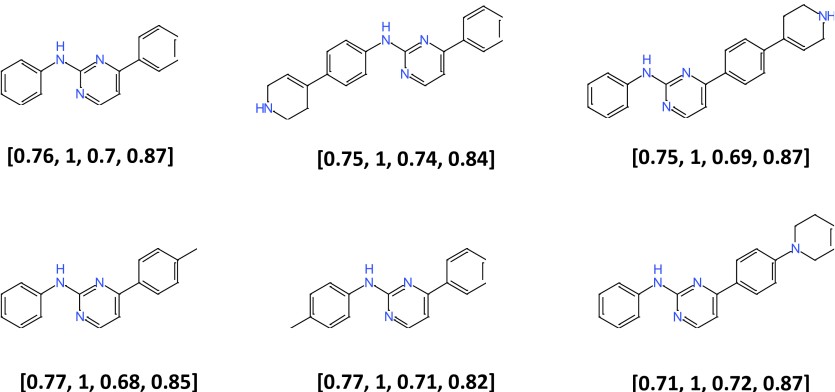

[0.76, 1, 0.7, 0.87]  [0.75, 1, 0.74, 0.84]  [0.75, 1, 0.69, 0.87]

[0.77, 1, 0.68, 0.85]  [0.77, 1, 0.71, 0.82]  [0.71, 1, 0.72, 0.87]

Figure 9: Generated molecules by InversionGNN. These four scores symbolize the values for QED, normalized SA, JNK3, and GSK3$\beta$, respectively.

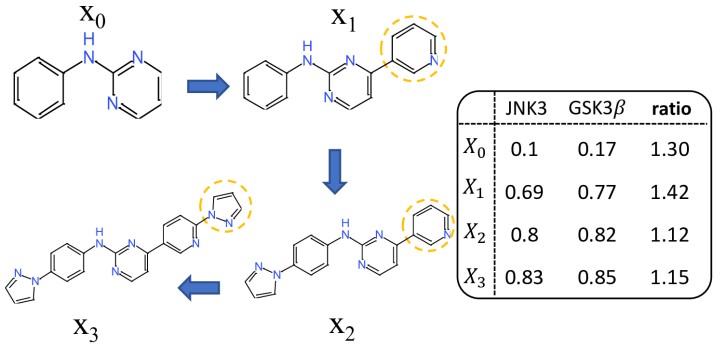

| | JNK3 | GSK3$\beta$ | ratio |
|---|---|---|---|
| $X_0$ | 0.1 | 0.17 | 1.30 |
| $X_1$ | 0.69 | 0.77 | 1.42 |
| $X_2$ | 0.8 | 0.82 | 1.12 |
| $X_3$ | 0.83 | 0.85 | 1.15 |

Figure 10: Generated molecules by I-LS, property scores and loss ratio.

