# OpenReview forum: "InversionGNN: A Dual Path Network for Multi-Property Molecular Optimization"
_ICLR.cc/2025/Conference — ICLR 2025 Poster_

### Official Review · Reviewer_GVk8 · 2024-10-30

**Soundness:** 2
**Presentation:** 2
**Contribution:** 2
**Rating:** 6
**Confidence:** 4

**Summary:**

The authors propose a molecular multi-objective optimization algorithm, inversionGNN, which proposes direct prediction paths and inversion generation paths to solve conflicting attribute optimization problems in chemical space. However, some issues need to be further addressed.

**Strengths:**

The method proposed by the authors, inversionGNN, a multi-objective molecular optimization method, is performed in a discrete space, which is different from the traditional latent space-based optimization methods and is methodologically innovative.

**Weaknesses:**

The authors' motivation for proposing the algorithm in their article by suggesting a conflict in the molecular properties is not specific enough. Also, in the article, it is suggested that the viewpoint of existing methods does not take into account the effect of substructure on molecular properties. Still, in the method introduction and experimental setup of inversionGNN, the authors do not address the effect of substructure on molecular properties either. At the same time, I don't quite understand what the significance of these tasks above the ablation experimental setup is. The authors should highlight the ablation experimental setup, what would be the impact if the current direct prediction path is not used, and what would be the extent of the impact on the current method if another proxy model is used.

**Questions:**

1. In the introduction, the authors suggest that substructures have an effect on molecular properties, so does inversionGNN take into account the effect of substructures on molecules?
2. in a multi-objective optimization task, how are the weights for multiple objectives balanced, or is the direct prediction path just viewed as a multi-classification task to bootstrap the Pareto front search?
3. what is the specific significance of the tasks designed by the authors in the ablation experiments, while the reader should be more interested in seeing how the direct prediction paths and inversion generation paths proposed by the authors affect the method.
4. Could the authors elaborate on how the surrogate oracle (used in Eq. 3) is trained and validated? Specifically, what types of data and target objectives are used for this model, and how is its accuracy assessed across diverse molecules?
5. The motivation for the thesis needs to be clear and is currently presented as too generalized. The flowchart of the methodology needs to be further improved so that the reader can understand the whole process.

---

> ### Comment · Reviewer_GVk8 · 2024-11-22
> **There was no response from the author**
>
> There was no response from the author

---

> > ### Author Response · Authors · 2024-11-22
> > **Thanks for the reminder**
> >
> > Thanks for the reminder, we will post our rebuttal in 24 hours.

---

> ### Author Response · Authors · 2024-11-23
> **Response to Reviewer GVk8 (part 1/2)**
>
> We sincerely appreciate your time and effort in reviewing our paper and providing constructive feedback. We would like to address your questions and concerns below.
>
> > **Weakness & Questions 5:** The authors' motivation for proposing the algorithm in their article by suggesting a conflict in the molecular properties is not specific enough. The flowchart of the methodology needs to be further improved so that the reader can understand the whole process.
>
> Thanks for the comment. We would like to further clarify our **motivation** and **contributions**:
>
> Our goal is to introduce the traditional gradient-based Pareto optimization to address the conflicting or correlated properties in discrete chemical space for multi-objective molecular optimization. To achieve this goal, we propose (1) a novel InversionGNN framework (allowing gradient-based optimization) and (2) the use of relaxation techniques to apply gradient-based Pareto optimization to discrete chemical structures. These two points complement each other to achieve this goal. Detailed explanations are as follows:
>
> * **InversionGNN framework.** (1) **Motivation:** Most existing works merely employ a chemical property predictor as a discriminator to filter high-performance molecules. They do not make full use of acquired chemical knowledge. In addition, it is unclear how to conduct gradient-based optimization on discrete chemical structures. (2) **Contribution:** We propose a novel dual-path InversionGNN for molecular optimization, which leverages the gradients from the acquired chemical knowledge to guide molecular optimization. It is effective and sample-efficient.
> * **Pareto optimization in the discrete chemical space.** (1) **Motivation:** Chemical properties often exhibit conflicting or correlated relationships, rather than being independent. Most existing works neglect this phenomenon, details are discussed in lines 38-52. The Pareto optimization technique is designed to solve conflicting or correlated objectives. However, most existing studies focus on continuous Euclidean space. It is still unclear how to address the multi-objective optimization problem in the discrete chemical space. (2) **Contribution:** We adopt the relaxation technique and first adapt gradient-based Pareto optimization to the discrete chemical space. We also provide theoretical guarantees in discrete chemical space.
>
> Thanks for your suggestion about the clarity of the flowchart. We have updated the flowchart in the revision. We add numbers and explanations to each step of InversionGNN.
>
>
>
> > **Weakness & Question 1**: In the article, it is suggested that the viewpoint of existing methods does not take into account the effect of substructure on molecular properties. In the introduction, the authors suggest that substructures affect molecular properties, so does inversionGNN take into account the effect of substructures on molecules?
>
> Thanks for raising this issue. Yes, we take into account the effect of substructures. We follow the common practice in molecular research and decompose the molecule into substructures. Then the molecule graph can be represented as a scaffolding tree (each node is a substructure). The GNN learns a function between the scaffolding tree and property scores, i.e. the effect of substructure on molecular properties.
>
>
> We would like to clarify that the substructure is not our contribution. Most existing works are based on substructure, such as GCPN, RationaleRL, MARS, DST, Graph-GA, etc. Our InversionGNN is also based on substructure. The key difference of InversionGNN is how to leverage the knowledge about "the effect of substructure on molecular properties". InversionGNN decodes the chemical knowledge stored in the GNN through gradients instead of traditional discriminators.

---

> ### Author Response · Authors · 2024-11-23
> **Response to Reviewer GVk8 (part 2/2)**
>
> > **Question 2**: In a multi-objective optimization task, how are the weights for multiple objectives balanced, or is the direct prediction path just viewed as a multi-classification task to bootstrap the Pareto front search?
>
> In Algorithm 1, the weight is a hyperparameter. It can be determined based on practical drug design needs. For example, for two conflicting properties $A$ and $B$, $A$ is more important than $B$ in practical drug design, thus we can set the weight of $A$ to be larger than that of B, such as $3:1$. InversionGNN can find the Pareto optimal molecules conditioned on desired weight.
>
> As mentioned in Question 1, the GNN learns a function between the scaffolding tree and property scores. This chemical knowledge is implicitly stored in GNN. In the direct prediction path, the molecule is fed to the GNN to calculate the objective function. In the inversion path, we calculate the non-dominating gradient from the objective function to guide the molecular optimization. **Therefore, without the direct path, we cannot calculate the gradient in the inversion path.** The whole process is similar to training a neural network with forward pass and backward pass, while we are training a molecule. We decode the chemical knowledge stored in the GNN through gradients.
>
> > **Question 3**: What is the specific significance of the tasks designed by the authors in the ablation experiments, while the reader should be more interested in seeing how the direct prediction paths and inversion generation paths proposed by the authors affect the method.
>
> Thanks for the comment. The significance of the ablation experiments is as follows:
>
> * The "Optimization Process" shows InversionGNN's capability to generate molecules with the desired weight. In practical drug design, InversionGNN can help design specific molecular properties for different diseases.  Additionally, this characteristic of InversionGNN can also help minimize drug side effects.
> * "Search Efficiency" shows that InversionGNN is sample-efficient. In practical drug discovery, conducting wet experiments to determine the properties of a molecule is very time-consuming and laborious. Therefore, the sample-efficient characteristic is very important for accelerating drug design.
> * "Conflicting and Correlated Objectives" shows InversionGNN's ability to find the Pareto front with extremely conflicting and correlated objectives. This ability is vital in practical drug discovery.
>
> Either the direct prediction path or inversion generation path can **Not** work independently for ablation study. We would like to clarify that the direct prediction path and inversion generation path are as a whole. A single path cannot work independently. As we mentioned in Question 2, InversionGNN trains the molecules. Just like training a neural network, the forward pass and the backward pass cannot work separately. That is the reason why we did not evaluate the role of each path separately in the ablation study.
>
> **The baseline I-LS can be regarded as an ablation study of the Inversion path.** In baseline I-LS, we change our inversion path to traditional gradient aggregation, as illustrated in algorithm 2. Our InversionGNN outperforms I-LS significantly, which shows the effectiveness of our inversion path. For the direct path, it is the inference of the GNN and cannot be replaced by other modules.
>
> > **Question 4**: Could the authors elaborate on how the surrogate oracle (used in Eq. 3) is trained and validated? Specifically, what types of data and target objectives are used for this model, and how is its accuracy assessed across diverse molecules?
>
> We appreciate your attention to detail. We train the model on the ZINC 250K dataset. We follow existing works and break all the ZINC molecules into substructures (including single rings and single atoms), count their frequencies, and include the substructures whose frequencies are higher than 1000 into the vocabulary set. The final vocabulary contains 82 substructures, including the frequent atoms like carbon atom, oxygen atom, nitrogen atom, and frequent rings like benzene ring. Then we clean the data by removing the molecules containing out-of-vocabulary substructure and having 195K molecules left. Then we randomly select 10K molecules for training the surrogate oracle. We adopt scaffolding tree as molecule representation (each node is a substructure). We randomly select 5K molecules as the validation set. We stop training when the validation performance does not change significantly for 10 epochs. The objective function is Binary Cross Entropy (BCE) loss. Finally, the surrogate oracle is frozen for molecular optimization.
>
> We would like to clarify that molecular optimization is not a classification task, thus we do not have an additional test set to evaluate its accuracy. The validation set is randomly selected from the ZINC dataset, which is diverse enough to ensure the quality of the model.

---

> > ### Comment · Reviewer_GVk8 · 2024-11-25
> >
> > The author addressed my main concern

---

### Official Review · Reviewer_QRSa · 2024-11-03

**Soundness:** 2
**Presentation:** 2
**Contribution:** 2
**Rating:** 6
**Confidence:** 2

**Summary:**

This paper introduces InversionGNN, a method for multi-objective molecular optimization. InversionGNN comprises two paths: a direct prediction path and an inversion generation path. In the direct prediction path, a graph neural network (GNN) is trained to incorporate chemical knowledge. In the inversion generation path, the discrete molecular optimization problem is reformulated as a locally differentiable Pareto optimization problem, allowing the model to capture complex multi-property relationships. Experimental results demonstrate the effectiveness of InversionGNN on multi-objective molecular optimization benchmarks.

**Strengths:**

- Comprehensive Literature Review: The authors present a thorough review of the relevant literature, situating their work well within the existing field.
- The paper is well-organized.
- Extensive Experiments: The authors conduct a wide range of experiments, supporting the robustness of their results.

**Weaknesses:**

- Clarity of Main Concepts: The core ideas are difficult to grasp, especially for those without a background in multi-objective optimization. I recommend adding more intuitive explanations at the start of Section 4 to clarify the fundamental concepts.
Missing Preliminaries on Differentiable Scaffolding Trees: The paper would benefit from an additional preliminary section on differentiable scaffolding trees to provide context for these methods.
- Unclear Terminology and Contribution: Key terms like “inversion generation path” and “dual-path” lack sufficient explanation. What exactly does "inversion" mean in this context, and how does it contribute to the model's novelty? Please clarify the role and purpose of the "dual-path" structure.
- Differentiation from Vanilla GCN: The distinction between the "Direct Prediction Path" and a standard Graph Convolutional Network (GCN) is unclear. Section 4.2 could be revised to explicitly highlight how this path diverges from a traditional GCN and its specific contributions to the model.
- Role of Relaxation Techniques: It is not clear how the relaxation techniques, which reformulate the discrete molecule Pareto optimization problem into a locally differentiable one, aid in capturing trade-offs among conflicting or correlated properties. Additional explanation on how this helps to balance multiple properties would improve understanding.

While this paper shows promise, I am not an expert in multi-objective optimization, and some core concepts remain unclear to me. I will finalize my score after reviewing feedback from another reviewer with more expertise in this area.

**Questions:**

See Weaknesses Section

---

> ### Author Response · Authors · 2024-11-25
> **Response to Reviewer QRSa (part 1/2)**
>
> Thank you for taking the time to review our InversionGNN and providing valuable feedback. We sincerely appreciate your efforts. We would like to address your questions and concerns below.
>
> > **Weakness 1**: Clarity of Main Concepts: The core ideas are difficult to grasp, especially for those without a background in multi-objective optimization. I recommend adding more intuitive explanations at the start of Section 4 to clarify the fundamental concepts. Missing Preliminaries on Differentiable Scaffolding Trees: The paper would benefit from an additional preliminary section on differentiable scaffolding trees to provide context for these methods.
>
> **We would like to illustrate the basic concept of multi-objective optimization with an example.** A classic example of Pareto optimality involves two individuals, $A$ and $B$, dividing $10$  dollars. Thus we have $A+B\leq10$. Any allocation where one person's situation cannot be improved without harming the other is Pareto optimal. For allocation $[A, B]$, $[2, 8]$ is Pareto optimal because an increase in $A$'s share would result in a decrease for $B$. Similarly, $[8, 2]$ is also Pareto optimal. However, $[1,8]$ is not Pareto optimal as we can increase $A$'s allocation without damaging $B$'s benefit. Therefore, the all possible Pareto optimal solutions are $[[0, 10],[1, 9],[2, 8],[3, 7],[4, 6],[5, 5],[6, 4],[7, 3],[8, 2],[9, 1],[10, 0]]$. In this example, objectives $A$ and $B$ are **conflicting**. It is impossible to maximize $A$ and $B$ simultaneously.
>
> In Pareto optimization, (1) one goal is to find Pareto optimal solutions conditioned on desired weight. For example, if we set the weight to $4:1$, then the ideal solution is $[8,2]$. (2) Another goal is to find all possible Pareto solutions. In this example, they are $[[0, 10],[1, 9],[2, 8],[3, 7],[4, 6],[5, 5],[6, 4],[7, 3],[8, 2],[9, 1],[10, 0]]$.
>
> In molecular optimization, if the two properties have conflicting relationships, it is infeasible to generate a single molecule that simultaneously maximizes all properties. Therefore, the goals of multi-objective molecular optimization are:
> * Find Pareto optimal molecules conditioned on desired weight.
> * Find all possible  Pareto optimal molecules.
>
> These two goals correspond to the two questions in our experiment section.
>
> **Thanks for your suggestions. We add the explanations at the start of Section 4 to clarify the fundamental concepts. And we add the preliminary of differentiable scaffolding tree in the revision.**
>
>
> > **Weakness 2**: Unclear Terminology and Contribution: Key terms like “inversion generation path” and “dual-path” lack sufficient explanation. What exactly does "inversion" mean in this context, and how does it contribute to the model's novelty? Please clarify the role and purpose of the "dual-path" structure.
>
> Thanks for the comment. We would like to illustrate the terminology as follows:
>
> **The explanation of the dual-path framework:** The surrogate GNN learns a function between the scaffolding tree and property scores. This chemical knowledge is implicitly stored in GNN.  In the direct prediction path, the molecule is fed to the GNN to calculate the objective function. In the inversion path, we calculate the non-dominating gradient w.r.t molecule structure from the objective function to guide molecular optimization. **The whole process is similar to training a neural network with forward pass and backward pass, while we are training a molecule.** To help understand, we have added numbers and explanations to Figure 1 in the revision.
>
>
> "Inversion" means the gradient is back-propagated to the molecule.  How does it contribute to the model's novelty? (1) It allows gradient-based optimization of molecules, thus we can incorporate gradient-based Pareto optimization with proper adaptation. (2) The gradient decodes implicit chemical knowledge stored in GNN, while traditional methods merely use GNN as a discriminator to filter molecules.

---

> ### Author Response · Authors · 2024-11-25
> **Response to Reviewer QRSa (part 2/2)**
>
> > **Weakness 3**: Differentiation from Vanilla GCN: The distinction between the "Direct Prediction Path" and a standard Graph Convolutional Network (GCN) is unclear. Section 4.2 could be revised to explicitly highlight how this path diverges from a traditional GCN and its specific contributions to the model.
>
> Thanks for your suggestions! We have revised Section 4.2 and added explanations about the difference between InversionGNN and GCN and its specific contributions.
>
> The key difference between InversionGNN and Vanilla GCN is the calculation process, but their network structures are the same. Vanilla GCN is often used for prediction tasks, and its inference process is the direct prediction path in InversionGNN. However, InversionGNN contains an additional inversion path for generation tasks, which allows the inverse of the gradients to the input molecules. Therefore, InversionGNN can train the input molecules with gradients, while Vanilla GCN cannot.
>
> > **Weakness 4**: Role of Relaxation Techniques: It is not clear how the relaxation techniques, which reformulate the discrete molecule Pareto optimization problem into a locally differentiable one, aid in capturing trade-offs among conflicting or correlated properties. Additional explanation on how this helps to balance multiple properties would improve understanding.
>
> Thanks for the comment. In mathematical optimization, relaxation is a modeling strategy. A relaxation is an approximation of a difficult problem by a nearby problem that is easier to solve. In chemical space, finding the global Pareto optimal molecules at one step is difficult. Therefore, we relax the problem to an iterative process. At each iteration, we try to find the local Pareto optimal molecules in the neighbor set. After a certain number of iterations, it gradually approaches the global Pareto optimal molecules. The whole process can be regarded as a greedy algorithm. This idea comes from the relaxation of finite-difference discretizations in solving differential equations.
>
>
> The direction of the Non-Dominating Gradient helps to balance multiple properties. $d_{nd}$ ensures the angle between $d_{nd}$ and the gradient $g_i$ of each objective is less than $\frac{\pi}{2}$. Even if there are conflicts between properties, it can ensure that each property will be improved in the optimization. Details are illustrated in Appendix B.

---

> ### Author Response · Authors · 2024-11-26
> **Looking forward to your reply**
>
> Dear Reviewer QRSa,
>
> Thank you for your thoughtful feedback and valuable suggestions to improve the clarity of our paper. We sincerely appreciate your efforts. As we have not yet received your response, we kindly request your feedback on whether our explanations have addressed your questions. We are more than willing to provide further clarifications or additional information to ensure that our contributions are clearly understood.
>
> Thank you once again for your valuable insights, and we look forward to your feedback!

---

> ### Author Response · Authors · 2024-12-02
> **The discussion period will end on December 2nd**
>
> Dear Reviewer QRSa,
>
> We sincerely thank you for providing valuable feedback. As the discussion period will end in one day, we would be grateful if you could allocate some time to review our responses.

---

> > ### Comment · Reviewer_QRSa · 2024-12-02
> >
> > Thanks for addressing my concerns. I raised my score to 6.

---

### Official Review · Reviewer_QPc3 · 2024-11-04

**Soundness:** 3
**Presentation:** 3
**Contribution:** 2
**Rating:** 6
**Confidence:** 3

**Summary:**

The paper presents the InversionGNN framework, which aims to optimize molecular properties for drug discovery using a dual-path GNN approach. It consists of a Direct Prediction Path, where a GNN is trained to predict multiple chemical properties, leveraging learned chemical knowledge to enhance understanding of optimal functional group combinations and an Inversion Generation Path that employs a gradient-based Pareto optimization method to generate molecules that meet specified properties, addressing challenges posed by conflicting or correlated objectives. Key contributions include the integration of property prediction knowledge into molecular optimization processes and a novel relaxation technique that adapts gradient-based Pareto optimization for discrete chemical spaces, facilitating the exploration of diverse Pareto-optimal solutions. Empirical evaluations show that InversionGNN is both effective and sample-efficient, outperforming existing methods in multi-objective drug discovery scenarios by successfully balancing trade-offs among various chemical properties.

**Strengths:**

①The introduction of a dual-path GNN effectively combines property prediction with molecular generation. This approach enhances the integration of learned chemical knowledge into the optimization process, allowing for more informed and effective molecular design.

②The method demonstrates significant sample efficiency in exploring the chemical space. By leveraging gradient-based Pareto optimization, InversionGNN can identify high-quality molecules with fewer oracle calls compared to traditional methods, making it a practical choice for real-world applications where computational resources may be limited.

③The paper includes extensive experimental evaluations across various real-world datasets and multi-objective settings. These experiments provide robust evidence of the method's effectiveness and its ability to generate diverse Pareto-optimal solutions, validating its utility in drug discovery scenarios.

④The theoretical analysis, including Theorem 4.2, offers insights into the convergence properties of InversionGNN. This foundation not only strengthens the claims made about the method's effectiveness but also provides a framework for future research in multi-objective molecular optimization.

**Weaknesses:**

①The assumptions underlying Theorem 4.2 are not clearly articulated or justified within the text. Possible assumptions that could be relevant include the continuity and differentiability of the objective functions, the boundedness of molecular sizes, and the stability of the optimization process under varying conditions. The absence of a discussion on these assumptions raises concerns about the robustness of the theoretical claims presented. Without a thorough examination, it becomes challenging to assess the practical applicability of the results in real-world scenarios. Furthermore, the theoretical framework predominantly focuses on the specific architecture and objectives of InversionGNN. This narrow focus limits the generalizability of the findings, as it does not consider how the proposed method may perform under different conditions or with various types of optimization problems. Expanding the theoretical discussion to include a broader range of contexts could enhance the credibility and utility of the results.

②The majority of experiments in this paper compare InversionGNN only with I-LS, which is a very simplistic linear scalarization model. Demonstrating that InversionGNN outperforms I-LS does not provide significant insights into its effectiveness or practical advantages. Additionally, many experiments only contrast InversionGNN with older methods, while comparisons with more recent advancements, such as HN-GFN, are limited to just two lightweight experiments. The primary experiments lack comprehensive comparisons with contemporary approaches, which could better contextualize the performance of InversionGNN and underscore its contributions to the field.

③The paper lacks thorough experimental evaluations against SOTA methods in multi-objective optimization. A detailed comparative analysis with current leading techniques would provide a clearer understanding of InversionGNN's strengths and weaknesses. By not including such comparisons, the paper misses an opportunity to establish a more compelling case for the novelty and superiority of InversionGNN in the context of existing literature. Including SOTA comparisons could significantly enhance the paper's contributions and relevance to researchers in the field.

**Questions:**

Why doesn't InversionGNN make an all-encompassing and detailed comparison with HN-GFN, but rather such a detailed comparison with I-LS.

---

> ### Author Response · Authors · 2024-11-25
> **Response to Reviewer QPc3 (part 1/2)**
>
> Thanks for your constructive comments! We try our best to address your concerns about theoretical findings and experiments. As suggested by the reviewer, extensive additional experiments are conducted to further support our contributions.
>
>
> >**Weakness 1.1**: The assumptions underlying Theorem 4.2 are not clearly articulated or justified within the text. Possible assumptions that could be relevant include the continuity and differentiability of the objective functions, the boundedness of molecular sizes, and the stability of the optimization process under varying conditions. The absence of a discussion on these assumptions raises concerns about the robustness of the theoretical claims presented. Without a thorough examination, it becomes challenging to assess the practical applicability of the results in real-world scenarios.
>
> Thanks for raising this concern. **We would like to clarify that the assumptions of Theorem 4.2 are justified in Appendix B.1 of the original submission.** Our assumption is reasonable and consistent with the general situation of drug discovery. We would like to provide a more detailed explanation of the assumptions you are concerned about:
> * The continuity and differentiability of the objective functions: We trained a Graph Convolutional Network (GCN) as the surrogate model and used Binary Cross Entropy (BCE) as the objective function. Both GCN and BCE are continuous and differentiable.
> * The boundedness of molecular sizes: The majority of drugs have molecular weights below 550 [1]. Additionally, druglikeness (QED) would decrease significantly when the molecule size is too large [2]. The boundedness is reasonable as most drugs are small molecules.
> * The stability of the optimization process under varying conditions: The assumption of submodularity is motivated by experimental observations. For instance, in the optimization process, we noticed that the objective values, such as QED, can increase rapidly with just adding a few substructures. However, as we added more substructures, the growth rate began to decrease when the property scores reached the upper bound. The curvature assumption is a generic condition of the submodularity.
>
>
> >**Weakness 1.2**: The theoretical framework predominantly focuses on the specific architecture and objectives of InversionGNN, which limits the generalizability of the findings.
>
> Thanks for the comment. **Our theoretical framework can also be generalized to other scenarios. However, we would like to clarify that the focus of our paper is the InversionGNN framework in molecular optimization.** We think that studying the theoretical properties of Pareto optimization in other scenarios will be interesting in future work. Here are detailed explanations:
>
> * The generalizability of the assumptions: Using a continuous and differentiable neural network to learn real-world data distribution is common practice.   In the real world, data generally has a bound, not infinite.
> * The generalizability of the non-dominating gradient $d_{nd}$: [3] discussed the theoretical properties of the non-dominating gradient. $d_{nd}$ converges to the Pareto optimal solution without any additional conditions (such as convexity). It only requires the function to be differentiable.
>
> When generalizing our theoretical framework to other scenarios, we should consider a new geometry of the objective landscape. It is a case-by-case factor that depends on the application scenario.
>
> **Reference:**
>
> [1] Chemical properties of antimicrobials and their uniqueness. Antibiotic discovery and development.
>
> [2] Quantifying the chemical beauty of drugs. Nature chemistry.
>
> [3] Multi-task learning with user preferences: Gradient descent with the controlled ascent in Pareto optimization. ICML 2020.

---

> ### Author Response · Authors · 2024-11-25
> **Response to Reviewer QPc3 (part 2/2)**
>
> >**Weakness 2 & Weakness 3 & Question 1**: The paper lacks thorough experimental evaluations against SOTA methods in multi-objective optimization, such as HN-GFN.
>
> Thanks for your suggestion. We agree that including such comparisons could significantly enhance the paper's contributions.  We add empirical comparisons with SOTA methods in multi-objective molecular optimization: HN-GFN  and MOGFN-AL. The experimental settings are consistent with Section 5.2. We extend the results as follows:
>
> * Molecular Optimization Conditioned on Desired Weight Task (Section 5.2 Q1)：
>
> | Method    | Nov($\uparrow$) | Div($\uparrow$) |  APS($\uparrow$) | NU($\downarrow$)  |
> | :---          |    :----: |    :----: | :----:     | :----:    |
> | MOGFN-PC      | **100%**      | 0.507 $\pm$   0.024    | 0.393 $\pm$  0.036      | 0.088  $\pm$  0.014   |
> | HN-GFN      | **100%**      | **0.571** $\pm$  0.032     | 0.418 $\pm$  0.022      | 0.072  $\pm$  0.015   |
> | I-LS          | **100%**      | 0.541 $\pm$  0.007      | 0.529 $\pm$  0.006      | 0.049 $\pm$  0.002     |
> | InversionGNN    | **100%**      | 0.435 $\pm$  0.009      | **0.648** $\pm$  0.012      | **0.026** $\pm$  0.001    |
>
> * Multi-Objective Drug Discovery (Section 5.2 Q2)
>
>  &ensp; &ensp; &ensp; &ensp;&ensp;&ensp; &ensp;&ensp; &ensp; &ensp; &ensp; &ensp; &ensp; &ensp; GSK3$\beta$ + JNK3    &ensp; &ensp;&ensp; &ensp;&ensp;  &ensp;&ensp; &ensp; &ensp; GSK3$\beta$ + JNK3 + QED + SA
> | Method    | Div($\uparrow$) |HV($\uparrow$) |  Div($\uparrow$) |HV($\uparrow$) |
> | :---          |    :----: |  :----: |:----: |  :----: |
> | MOGFN-AL    |0.694 $\pm$ 0.012 |0.567 $\pm$  0.057    | 0.706 $\pm$ 0.009 |0.377 $\pm$  0.046   |
> | HN-GFN   |**0.794** $\pm$ 0.006 |0.592 $\pm$  0.042      |  0.732 $\pm$ 0.005|0.374 $\pm$  0.039     |
> | I-LS     |0.685 $\pm$ 0.004|0.475 $\pm$  0.028      |0.712 $\pm$ 0.007 |0.308 $\pm$  0.022     |
> | InversionGNN  | 0.775 $\pm$ 0.008| **0.763** $\pm$ 0.031 | **0.765** $\pm$ 0.006 |**0.519** $\pm$ 0.038 |
>
> We will include the results in the revised version, **including new visualization results.**  We will update the revision in the next few hours. We apologize for the late submission of the rebuttal.

---

> > ### Comment · Reviewer_QPc3 · 2024-11-25
> >
> > Dear Authors,
> >
> > Thank you for your efforts in addressing my comments by supplementing additional baselines and main experiments. However, I have noticed several issues that remain unresolved:
> >
> > * While new experimental results have been provided, there is a lack of analysis or discussion regarding these results, which makes it difficult to understand their implications or how they contribute to the overall findings.
> > * The newly added experiments do not align with the structure or presentation of the main experiment table, resulting in inconsistencies that hinder comparison and clarity.
> > * The updated experiments and supplementary information have not been incorporated into the manuscript, leaving the current version incomplete and not reflective of your revisions.
> >
> > If these issues are properly addressed, including a thorough analysis of the results, consistency in presentation, and integration into the manuscript, I would be open to reconsidering my evaluation and potentially increasing the score. At this stage, however, I will maintain my original rating.

---

> > > ### Author Response · Authors · 2024-11-26
> > > **We have updated the manuscript**
> > >
> > > Dear Reviewer QPc3,
> > >
> > > Thank you once again for your patience and understanding. We have updated our manuscript and addressed the following issues in the revision:
> > > * We provide detailed analysis regarding the new results.
> > > * We incorporate the newly added results into the existing table.
> > > * We include a thorough analysis of InversionGNN and HN-GFN in Section 5.2.
> > >
> > > We appreciate your time engaging in the discussion. We would like to address any further questions you may have.

---

> > > > ### Author Response · Authors · 2024-11-28
> > > > **Looking forward to your reply**
> > > >
> > > > Dear Reviewer QPc3,
> > > >
> > > > Thanks for your contributions to the reviewing process. As the deadline for the discussion approaches, we kindly request your feedback on whether our revision has addressed your concerns. We would be more than willing to engage in further discussions and make any necessary improvements.

---

> ### Author Response · Authors · 2024-11-29
> **Looking forward to your feedback**
>
> Dear Reviewer QPc3,
>
> Thank you very much for your time and consideration. As the discussion period is nearing its end, we would appreciate it if you could confirm whether our revisions have addressed your concerns. If there are any remaining issues, we are willing to engage in further discussion and make additional adjustments.

---

> > ### Author Response · Authors · 2024-12-02
> > **The discussion period will end on December 2nd**
> >
> > Dear Reviewer QPc3,
> >
> > We sincerely thank you for providing valuable feedback.  As the discussion period will end in one day, we would be grateful if you could allocate some time to review our revisions.

---

> > > ### Comment · Reviewer_QPc3 · 2024-12-02
> > >
> > > I have reviewed the revised manuscript and adjusted my score based on the improvements you have made.
> > >
> > > However, I want to emphasize that repeatedly urging reviewers for feedback will not influence the score. Reviewers base their evaluations solely on the quality of the manuscript, not on the frequency of follow-up emails. I kindly ask that you refrain from such behavior in the future.

---

> > > > ### Author Response · Authors · 2024-12-02
> > > >
> > > > We sincerely apologize for any inconvenience caused by our previous follow-up emails. We appreciate your feedback and will ensure such behavior does not occur in the future. Thank you for your understanding.

---

### Official Review · Reviewer_ToHW · 2024-11-04

**Soundness:** 4
**Presentation:** 3
**Contribution:** 3
**Rating:** 8
**Confidence:** 4

**Summary:**

The article introduces InversionGNN, a dual-path graph neural network designed for multi-objective molecular optimization in drug discovery, addressing the challenge of balancing conflicting molecular properties. It  combines a direct prediction path, which learns chemical relationships between substructure and properties, with an inversion generation path that uses gradient-based Pareto optimization to generate molecules achieving trade-offs. The authors demonstrate the effectiveness of InversionGNN by empirical analysis on synthetic tasks and multi-objective drug discovery.

**Strengths:**

- The paper addresses an important problem, and the overall presentation is good, including writing and visualizations.

- The proposed gradient-based Pareto inversion module aligns with the intuition of multi-objective optimization (MOO) and offers strong interpretability.

- The experimental analysis is thorough, validating the model’s extensive exploration of the Pareto front in MOO from multiple perspectives, not limited to the molecular domain. Additionally, the ablation study provides interesting insights, such as showing that InversionGNN's incremental addition of substructures aligns better with weighted vectors.

**Weaknesses:**

- My main concern is the claimed superiority of InversionGNN in molecular design results. On close examination of Table 1, it appears that DST, the most competitive baseline, performs nearly on par with InversionGNN across various metrics. Since the primary enhancement in InversionGNN is the gradient-based Pareto inversion module, this raises the question of whether the impact of this module is limited and whether the tokenization approach of the scaffolding tree is actually more critical for molecular design.

- The time complexity analysis is insufficient. In practical applications, QP-solving complexity can be significant. While the paper suggests that computing the non-dominating gradient is negligible in theory, I recommend adding a comparison of computation times across methods to provide clearer insights.

- The motivation behind oracle budget selection is not well-explained. What was the rationale for the oracle call budgets chosen for each baseline? The authors should consider controlling for a consistent budget across models to ensure fair supervision signals, which would also facilitate comparative analysis.

- The definition of the neighborhood set is somewhat ambiguous. How is the neighborhood set specifically defined in the context of molecular design?

**Questions:**

Please refer to the questions mentioned in the Weaknesses.

---

> ### Author Response · Authors · 2024-11-25
> **Response to Reviewer ToHW (part 1/2)**
>
> Thank you for your valuable feedback on our paper. We appreciate your time and effort in reviewing our work. We would like to address your questions and concerns below.
>
> >**Weakness 1**: My main concern is the claimed superiority of InversionGNN in molecular design results. On close examination of Table 1, it appears that DST, the most competitive baseline, performs nearly on par with InversionGNN across various metrics. Since the primary enhancement in InversionGNN is the gradient-based Pareto inversion module, this raises the question of whether the impact of this module is limited and whether the tokenization approach of the scaffolding tree is actually more critical for molecular design.
>
> Thanks for raising this concern. We would like to clarify that the gradient-based Pareto inversion module is critical for dealing with conflicting or correlated properties. We agree that the scaffolding tree plays an important role in molecular design, yet it does not contribute to dealing with multiple properties.
>
>
> The metrics in Table 1 are commonly used in molecular optimization. For a fair comparison, we follow the traditional metrics. However, they cannot well measure the performance of Pareto optimality in multi-objective setting. For example, APS focuses more on the average score but ignores the following two aspects: (1) whether the generated molecules are conditioned on the desired weight; and (2) whether all possible Pareto optimal molecules are generated. The two aspects are evaluated as follows:
>
> (1) Non-Uniformity (NU) in Table 3 measures whether the generated molecules are conditioned on the desired weight. It shows that InversionGNN outperforms existing baselines by a large margin. Figure 5 also shows the InversionGNN's ability to generate Pareto optimal molecules conditioned on the desired weight.
>
> (2) As suggested by Reviewer QPc3, we add the metric HyperVolume (HV) to evaluate whether all possible Pareto optimal molecules are generated. HV measures the objective space spanned by a set of non-dominated solutions. The results are as follows：
>
>
>
>  &ensp; &ensp; &ensp; &ensp;&ensp;&ensp; &ensp;&ensp; &ensp; &ensp; &ensp; &ensp; &ensp; &ensp; GSK3$\beta$ + JNK3    &ensp; &ensp;&ensp; &ensp;&ensp;  &ensp;&ensp; &ensp; &ensp; GSK3$\beta$ + JNK3 + QED + SA
> | Method    | Div($\uparrow$) |HV($\uparrow$) |  Div($\uparrow$) |HV($\uparrow$) |
> | :---          |    :----: |  :----: |:----: |  :----: |
> | MOGFN-AL    |0.694 $\pm$ 0.012 |0.567 $\pm$  0.057    | 0.706 $\pm$ 0.009 |0.377 $\pm$  0.046   |
> | HN-GFN   |**0.794** $\pm$ 0.006 |0.592 $\pm$  0.042      |  0.732 $\pm$ 0.005|0.374 $\pm$  0.039     |
> | I-LS     |0.685 $\pm$ 0.004|0.475 $\pm$  0.028      |0.712 $\pm$ 0.007 |0.308 $\pm$  0.022     |
> | InversionGNN  | 0.775 $\pm$ 0.008| **0.763** $\pm$ 0.031 | **0.765** $\pm$ 0.006 |**0.519** $\pm$ 0.038 |
>
> >**Weakness 2**: The time complexity analysis is insufficient. In practical applications, QP-solving complexity can be significant. While the paper suggests that computing the non-dominating gradient is negligible in theory, I recommend adding a comparison of computation times across methods to provide clearer insights.
>
> We solve $K=5000$ QP problem per iteration. The average wall clock time of each iteration in InversionGNN is 3.28s. The molecular graph is usually small graph, and the dimension of quadratic programs is low and does not significantly increase the complexity.
>
> The wall clock times of each method are closely related to their hyperparameter settings. For a fair comparison, we divide the current methods for dealing with multiple objectives in molecular optimization into InversionGNN, Genetic algorithm, and Bayesian optimization, as illustrated in the Introduction Section. We provide a theoretical analysis of their computational bottlenecks for each method during **one iteration**:
>
> * InversionGNN. If we perform $K$ gradient descent steps in each iteration, the time complexity is $O(K(m^2n+m^3))$, where $n$ is the dimension of the gradients and $m$ is the number of objectives, usually $n\gg m$.
> * Genetic algorithm.  The computational bottleneck lies in the non-dominated sorting, which has a time complexity of $O(mp^2)$, where $p$ is the population size and $m$ is the number of objectives.
> * Bayesian optimization. The computational bottleneck lies in training the Gaussian process, which has a time complexity of $O(q^3)$, where $q$ is the number of training data points.

---

> ### Author Response · Authors · 2024-11-25
> **Response to Reviewer ToHW (part 2/2)**
>
> >**Weakness 3**: The motivation behind Oracle budget selection is not well-explained. What was the rationale for the Oracle call budgets chosen for each baseline? The authors should consider controlling for a consistent budget across models to ensure fair supervision signals, which would also facilitate comparative analysis.
>
> Thanks for the comment. **We set the Oracle budget according to the original setting of each method. It is the minimum oracle budget required for each method to converge.** Some methods, such as GCPN and MolDQN, require a large number of oracle calls to obtain satisfactory results. In contrast, methods, such as MARS, require less Oracle budget to converge, and increasing the Oracle budget does not significantly increase the performance. Therefore, it is difficult to find a consistent oracle budget suitable for all methods. **The Oracle budget is a vital indicator to measure whether a method is sample-efficient, and it is not just a hyperparameter.** And our InversionGNN outperforms all baselines by a large margin with the smallest oracle budget.
>
> We agree that controlling for a consistent budget across models ensures fair supervision signals, so we add a curve of different methods under the same budget. **We will include this curve in the revision and will update the revision in the next few hours. We apologize for the late submission of the rebuttal.**
>
> >**Weakness 4**: The definition of the neighborhood set is somewhat ambiguous. How is the neighborhood set specifically defined in the context of molecular design?
>
> The neighborhood set of molecule $x$, denoted $\mathcal{N}(x)$, is the set of all the possible molecules obtained by (1) imposing **one** local editing operation (add, delete or replace one substructure) to scaffolding tree $\mathcal{T}_x$ and (2) assembling the edited trees into molecules.

---

> > ### Comment · Reviewer_ToHW · 2024-11-26
> >
> > Thank you for your efforts in addressing my comments and most of the concerns are addressed. If the authors have added the promised experiment in the revised manuscript with sufficiently convincing results, I may reconsider updating my score.

---

> > > ### Author Response · Authors · 2024-11-26
> > > **We will submit the revision in a few hours**
> > >
> > > Dear Reviewer ToHW:
> > >
> > > Thank you for your patience! We are working on the revisions and are almost done. We will update as soon as the revisions are finished.

---

> ### Author Response · Authors · 2024-11-26
> **We have updated the manuscript**
>
> Dear Reviewer ToHW,
>
> Thank you once again for your time and patience. We have updated our manuscript and addressed the following issues in the revision:
> * We add additional results to support our claim in Section 5.2 (Weakness 1).
> * We include the computation complexity comparison in Appendix C (weakness 2).
> * We add a comparison for a consistent budget across models in Appendix E.3 (weakness 3).
> * We include the definition of the neighborhood set in Section 4.3 (weakness 4).
>
> We appreciate your time engaging in the discussion. We would like to address any further questions you may have.

---

> > ### Author Response · Authors · 2024-11-28
> > **Looking forward to your reply**
> >
> > Dear Reviewer ToHW,
> >
> > Thanks for your contributions to the reviewing process. As the deadline for the discussion approaches, we kindly request your feedback on whether our revision has addressed your concerns.  We would be more than willing to engage in further discussions and make any necessary improvements.

---

> > > ### Comment · Reviewer_ToHW · 2024-11-28
> > >
> > > Thanks for your revisions and efforts. I believe the authors have addressed most of the questions I initially raised. My last concern is that the additional experimental metrics, HV,  provided for Weakness 1 do not evaluate the performance of DST. I suggest that the authors include a comparison with this baseline in future versions, as it is highly relevant and competitive to the proposed method.
> > >
> > > Taking into account the authors' responses, the additional experiments and provided revisions, I have decided to update my rating to 8. Good luck.

---

> ### Author Response · Authors · 2024-11-28
>
> We will include the hypervolume of DST in Table 4 in the final version. Thank you again for your time and efforts!

---

### Author Response · Authors · 2024-11-26
**Summary of Revisions**

We sincerely thank all the reviewers for their insightful reviews and valuable comments, which are instructive for improving our paper further. We are glad that the reviewers appreciated the significance of our problem (ToHW, QPc3), the interest and novelty of our proposed framework (ToHW, QPc3, QRSa, GVk8), the soundness of our theoretical analysis (QPc3), the comprehensiveness of our experiments (ToHW, QPc3, QRSa), and the overall quality of our paper's writing (ToHW). We apologize again for the late submission of the rebuttal and thank you for your patience!


The reviewers also raised insightful and constructive concerns. We have made every effort to address all the concerns by providing sufficient evidence and requested results. Here is the summary of the major revisions:
* Add computation complexity comparison in Appendix C (Reviewer ToHW).
* Add additional comparison for a consistent budget across models in Appendix E.3 (Reviewer ToHW).
* Add the definition of the neighborhood set in Section 4.3 (Reviewer ToHW).
* Add comparison with recent multi-objective molecular optimization works in Section 5.2. We include additional results in Table 4 and Figure 4 (Reviewer QPc3).
* Add intuitive explanations in Section 4 to clarify the fundamental concepts of multi-objective molecular optimization (Reviewer QRSa).
* Add preliminary on differentiable scaffolding tree in Section 3 (Reviewer QRSa).
* Clarify the difference between InversionGNN and  Vanilla GCN in Section 4.2 (Reviewer QRSa).
* Update the flowchart of InversionGNN in Figure 1 (Reviewer GVk8).

The valuable suggestions from reviewers are very helpful for us to improve our paper.  **All the changes we've made in our revision are highlighted in blue.** We'd be very happy to answer any further questions.

---

### Author Response · Authors · 2024-11-27
**We welcome comments on our revision**

Dear Reviewers,

Thank you for your valuable feedback on our manuscript. We have made revisions based on your suggestions. As the revision deadline is approaching, we kindly request your feedback on whether our revision has addressed your concerns. If you have any further comments or suggestions, we would be more than willing to address them before the deadline.

Thank you once again for dedicating your valuable time to reviewing our work.

Best regards,

The Authors

---

### Meta-Review · Area_Chair_6133 · 2024-12-21

**Metareview:**

This paper proposes a gradient-based Pareto inversion method for multi-objective optimization of molecules based on differentiable molecular representation (differentiable scaffolding tree). The reviewers generally agree on novelty and strength of the work. I believe the experiments should include more strong baselines for single-objective molecule optimization with linear scalarization, e.g., [1], but this is not enough to reject the paper.

Overall, I recommend acceptance of the paper.

[1] Genetic-guided GFlowNets for Sample Efficient Molecular Optimization

**Additional Comments On Reviewer Discussion:**

All the reviewers agree on acceptance of the paper.

---

### Decision · Program_Chairs · 2025-01-22

Accept (Poster)